

**An improved perspective in the representation of soil moisture: potential added value of**

**SMOS disaggregated 1 km resolution product**

Samiro Khodayar[1], Amparo Coll[2], Ernesto Lopez-Baeza[2]

[1] Institute of Meteorology and Climate Research (IMK-TRO), Karlsruhe Institute of

Technology (KIT), Karlsruhe, Germany

[2] University of Valencia, Spain. Earth Physics and Thermodynamics Department. Climatology

from  Satellites Group

* Corresponding author. E-mail address: samiro.khodayar@kit.edu (S. Khodayar)

Institute for Meteorology and Climate Research, Karlsruhe Institute of Technology (KIT),

Postfach 3640, 76021 Karlsruhe, Germany





**Abstract**
This study uses the synergy of multiresolution soil moisture (SM) satellite estimates from the
Soil Moisture Ocean Salinity (SMOS) mission, a dense network of ground-based SM
measurements, and a Soil Vegetation Atmosphere Transfer (SVAT) model, SURFEX
(Externalized Surface) – module ISBA (Interactions between Soil-Biosphere-Atmosphere), to
examine, i) the comparison and suitability of different operational SMOS SM products to
provide realistic information on the water content of the soil as well as the added value of the
newly released SMOS Level 4 3.0 "all weather" disaggregated ~ 1 km SM (SMOS_L4$^{3.0}$),
and ii) its potential impact for improving uncertainty associated to SM initialization in land
surface modelling. Three different data products from SMOS-L3 (~ 25 km), L2 (~15 km), and
disaggregated L4 3.0 (~1km) are investigated. In situ SM observations over the Valencia
Anchor Station (VAS; SMOS Calibration/Validation (Cal/Val) site in Europe) are used for
comparison. The SURFEX-ISBA model is used to simulate point-scale surface SM (SSM)
and, in combination with high-quality atmospheric information data, namely ECMWF and the
SAFRAN meteorological analysis system, to obtain a representative SSM mapping over the
VAS. The sensitivity to SSM initialization, particularly to realistic initialization with
SMOS_L4$^{3.0}$ to simulate the spatial and temporal distribution of SSM is assessed. Results
demonstrate: (a) all SMOS products correctly capture the temporal patterns, but, the spatial
patterns are not accurately reproduced by the coarser resolutions probably in relation to the
contrast with point-scale in situ measurements. (b) The potential of SMOS-L4$^{3.0}$ product is
pointed out to adequately characterize SM spatio-temporal variability reflecting patterns
consistent with intensive point scale SSM samples on a daily time scale. The restricted
temporal availability of this product dictated by the revisit period of the SMOS satellite
compromises the averaged SSM representation for longer periods than a day. (c) A seasonal





analysis points out improved consistency during December-January-February and September-
October-November in contrast to significantly worse correlations in March-April-May (in
relation to the growing vegetation) and June-July-August (in relation to low SSM values $< 0.1$
$m^3/m^3$ and low spatial variability). (d) Perturbation simulations with the SURFEX-ISBA
SVAT (Soil-Vegetation-Atmosphere Transfer) model demonstrate the impact of the initial
SSM scenarios on its temporal evolution. (e) The combined use of the SURFEX-ISBA SVAT
model with the SAFRAN system, initialized with SMOS-L4$^{3.0}$ 1 km disaggregated data is
proven to be a suitable tool to produce regional SM maps with high accuracy which could be
used as initial conditions for model simulations, flood forecasting, crop monitoring and crop
development strategies, among others.
*Key Words: soil moisture, SMOS 1-km disaggregated product, SURFEX, Valencia Anchor*
*Station, realistic initialization, SAFRAN*



## 1. Introduction

Reliability of climate and hydrological models is constrained by associated uncertainties, such as input parameters. Among them, soil moisture is a variable of pivotal importance controlling the exchanges of water and energy at the surface/atmosphere interface (Entekhabi et al., 1996). Thus, it is a highly relevant variable for climate, hydrology, meteorology and related disciplines (e.g. Seneviratne et al. 2010).

Soil moisture is greatly variable spatially, temporally and across scales. The spatial heterogeneity of soil, vegetation, topography, land cover, rainfall and evapotranspiration are accounted responsible (Western et al., 2002; Bosh et al., 2007). An adequate representation of the high spatio-temporal variability of soil moisture is needed to improve climate and hydrological modelling (Koster et al., 2004; Seneviratne et al., 2006; Brocca et al., 2010). Its impact has been seen on time scales from hours to years (e.g., ~ 20 km scale: Taylor and Lebel, 1998; droughts: Schubert et al., 2004; decadal drying of the Sahel: Walker and Rowntree, 1977; hot extremes: Seneviratne et al., 2006b; Hirschi et al., 2011; decadal simulations: Khodayar et al., 2014). To obtain an appropriate representation of this variable, especially at high-resolution, is not an easy task mainly because of its high variability. Methods for the estimation of soil moisture can be divided in three main categories, (i) measurement of soil moisture in the field, (ii) estimation via simulation models, and (iii) measurement using remote sensing. In general, in situ measurements are far from global (e.g., Robock et al. 2000), and model simulations present important biases. Therefore, we have to rely on space-borne sensors to provide such measurements, but until recent times no dedicated, long-term, moisture space mission was attempted (Kerr, 2007).

Nowadays, by means of remote sensing technology surface soil moisture is available at global scale (Wigneron et al., 2003). The best estimations result from microwave remote sensing at low frequencies (Kerr, 2007; Jones et al., 2011). The SMOS (Soil Moisture and Ocean



Salinity; Kerr et al., 2001) mission is the first space-borne passive L-band microwave (1.4
GHz) radiometer measuring at low frequency soil moisture over continental surfaces as well
as ocean salinity (Kerr et al., 2001, 2010). SMOS delivers global surface soil moisture
measurements (~ 0-5 cm depth) at 0600 a.m. and 0600 p.m. LT (local time) in less than 3-
days revisit at a spatial resolution of ~ 44 km. The benchmark of the mission is to reach
accuracy better than 0.04 $m^3/m^3$ for the provided global maps of soil moisture (Kerr et al.,

78    2001).

SMOS data is not exempt of biases. Validating remote sensing-derived soil moisture products
is difficult, e.g. due to scale differences between the satellite footprints and the point
measurements on the ground (Cosh et al., 2004). However, in the last years a huge effort has
been made to validate the SMOS algorithm and its associated products. With this purpose, in
situ measurements across a range of climate regions were used assessing the reliability and
accuracy of these products using independent measurements (Delwart et al., 2008; Juglea et
al., 2010; Bircher et al., 2012; Dente et al., 2012; Gherboudj et al., 2012; Sánchez et al., 2012;
Wigneron et al., 2012). The strategy adapted by the European Space Agency (ESA) was to
develop specific land product validation activities over well-equipped monitoring sites. An
example for this is the Valencia Anchor Station (VAS; Lopez-Baeza et al., 2005a) in eastern
Spain, which was chosen as one of the two main test sites in Europe for the SMOS
Calibration/Validation (Cal/Val) activities. The validation sites were chosen to be slightly
larger than the actual pixel (3dB footprint), thus, VAS covers a 50x50 $km^2$ area. Within this
area, a limited number of ground stations were installed relying on spatialized soil moisture
information using the SVAT (Soil Vegetation Atmospheric Transfer) SURFEX (Externalized
Surface) model. Worldwide validation results reveal a coefficient of determination ($R^2$) of
about 0.49 when comparing the ~5 cm in situ soil moisture averages and the SMOS soil
moisture level 2 (SMOS-L2 ~ 15 km). For example, validation results by Bircher et al. (2012)





in Western Denmark show $R^2$ of 0.49-0.67 (SMOS retrieved initial soil moisture) and 0.97
(SMOS retrieved initial temperature). Besides, a significant under-/over-representation of the
network data (biases of $-$ 0.092-0.057 m³/m³) is also found. Over the Maqu (China) and the
Twente (The Netherlands) regions, the validation analysis resulted in $R^2$ of 0.55 and 0.51,
respectively, for the ascending pass observations, and of 0.24 and 0.41, for the descending
pass observations. Furthermore, Dente et al. (2012) pointed out a systematic SMOS soil
moisture  (ascending pass observations) dry bias of about 0.13 m³/m³ for the Maqu region and
0.17 m³/m³ for the Twente region. Validation of the SMOS level 3 product (SMOS-L3 ~ 35
km) shows that the general dry bias in SMOS-L2 is also present in SMOS-L3 SM.  This bias
is markedly present in the ascending products and shorter time series as described in Sanchez
et al. (2012) and Gonzalez-Zamora et al. (2015). In this case, the presence of dense vegetation
is seen to increase RMSE scores, whereas in low vegetated areas a lower dry bias is found
(Louvet et al. 2015).
Since the launch of the SMOS satellite, the processing prototypes of the SMOS L2 soil
moisture have evolved, and their quality has improved. Furthermore, efforts have been made
to cover the need of a reliable product with finer resolution for hydrological and climatic
studies where the spatial variability of soil moisture plays a crucial role, e.g. in the estimation
of land surface fluxes (evapotranspiration (ET) and runoff). Piles et al. (2011) presented a
downscaling approach to optimally combine SMOS´ soil moisture estimates with MODIS
visible/infrared (VIS/IR) satellite data into 1 km soil moisture maps over the Iberian
Peninsula (IP) without significant degradation of the root mean square error (RMSE). This
product has been evaluated using the REMEDHUS (REd de MEDicion de la HUmedad del
Suelo) soil moisture network in the semi-arid area of the Duero basin, Zamora, Spain (Piles et
al. 2014). Results show that downscaling maintains temporal correlation and root mean
squared differences with ground-based measurements, hence, capturing the soil moisture



dynamics. A big limitation for this downscaling approach is the lack of information in cloudy
conditions, which significantly limits the availability and usefulness of this product. Trying to
tackle this problem, a new product, SMOS Level 4 3.0 "all weather" disaggregated ~ 1 km
SM (SMOS_L4$^{3.0}$, the previous product is hereafter named SMOS_L4$^{2.0}$ ) was developed, in
which the limitation on clouds is taken into account and has been recently made available by
SMOS-BEC (Barcelona Expertise Centre).
Up to now, SMOS-L3 and -L2 products have extensively been validated as described above
and used for assimilation purposes in models (e.g. De Lannoy et al. 2016; Leroux et al. 2016);
however, few studies deal with the disaggregated 1 km SMOS-L4$^{0.2}$ and SMOS-L4$^{0.3}$ products
(mostly in relation to wildfire activity). In this study, the synergy of satellite reprocessed
SMOS soil moisture data obtained with improved processors, model simulations with the
SVAT SURFEX-ISBA and in situ stations from the VAS soil moisture network are used for
evaluation of the soil moisture fields. The first objective of this paper is to provide
information about the advantages and drawbacks of the different data sets and to assess the
added value of the SMOS-L4$^{3.0}$ product with respect to coarser resolution products. The
second objective is devoted to apply a methodology to derive soil moisture maps over the
VAS area to evaluate the usefulness of the SMOS-L4$^{3.0}$ product regarding future applications
such as realistic initialization in model simulations to reduce associated uncertainty. The
proposed investigation covers a one year period (a complete hydrological cycle) and focuses
on the semi-arid VAS area and the IP where water availability and fire risk are big
environmental issues, thus, knowledge of soil moisture conditions is of pivotal importance.
Furthermore, as spring time soil moisture anomalies over the IP are believed to be a pre-
cursor to droughts and heat waves in Europa (Vautard et al. 2007; Zampieri et al. 2009),
accurate monitoring and prediction of surface states in this region may be key for
improvements in seasonal forecasting systems.



The following objectives are then pursued: (a) Examination of soil moisture temporal and
spatial distribution with SMOS-derived soil moisture products over the investigation domain
using a multi-resolution approach: L3 (~ 25 km), L2 (~15 km), and L4$^{3.0}$ (~ 1 km), (b)
Validation with the in situ soil moisture measurements' network (VAS) to estimate the
reliability of the SMOS SM products, (c) Evaluation of the usefulness at different resolutions
and the added value of the 1 km product, (d) Modelization of point-scale soil moisture with
SURFEX-ISBA and spatialization over the VAS area using ground measurements for
verification, (e) Evaluation of the impact of realistic SM initialization using SMOS-L4$^{3.0}$ on
point-scale and regional model simulations over the VAS area. This investigation is structured
as follows, in Section 2, the study area and the data sets are presented including the ground
measurements, the SMOS data products, and the SURFEX-ISBA model and related
atmospheric forcings used. Section 3 summarizes the methodology applied. The results are
discussed in Section 4. Finally, conclusions are drawn in Section 5.

## 2. Study area and data set

2.1 Investigation domain and in situ measurements over the VAS
The main investigation areas in this study are the Iberian Peninsula and the Valencia Anchor
Station (VAS) site located in eastern Spain (39.69°-39.22° N,-1.7°-(-1.11°) W). The VAS site
covering approximately a 50x50 km$^2$ area was established in December 2001 by the
University of Valencia as a Calibration/Validation (Cal/Val) site for different low-resolution
Earth Observation data products (Bolle et al., 2006). The extension and homogeneity of the
area as well as the mostly flat conditions (slopes lower than 2%) make it an ideal reference
site. Nevertheless, the small variations in the area, 750 to 950 m, influence the climate of the
region, which oscillates between semiarid to dry-sub-humid. Most of the area is dedicated to



vineyards (65%), followed by trees, shrubs, forest and industrial and urban cover types.
Mostly bare soil conditions are observed beside the vineyard growing season (March/April to
September/October). Mean temperatures in the region are between 12°C and 14°C with
annual mean precipitation about 450 mm, with maximums in spring and autumn. Within the
VAS, a network consisting of eight ThetaProbe ML2x soil moisture stations was deployed by
the Climatology from Satellites Group from the Earth Physics and Thermodynamics
Department at the University of Valencia. The eight in situ stations are distributed over a
10x10 km$^2$ area (Figure 1), according to land use, soil type, and other environmental
conditions. Details about the characteristics of each station are summarized in Table 1. Soil
moisture measurements every 10 min, mostly from 2006, were carried out for the top first 5
cm. More details about the VAS characteristics and soil moisture measurements could be
found in Juglea et al. (2010). Precipitation measurements over the IP and the VAS are from
the AEMET (Agencia Estatal de Meteorología; Spanish Weather Service) network.
Measurements every 10 min are available.
2.2 The SMOS surface soil moisture products
ESA's derived SMOS Soil Moisture Level 2 (SMOS-L2) data product, ~ 15 km, contains the
retrieved soil moisture and optical thickness and complementary parameters such as
atmospheric water vapour content, radio frequency interferences and other flags. The SMOS-
L2 algorithms have been refined since the launch of SMOS, resulting in more precise SM
retrievals (ARRAY, 2014). The Level 3 SM product, SMOS-L3, was obtained from the
operational CATDS archive. This is a daily product that contains filtered data. The best
estimation of SM is selected for each node when several multi-orbit retrievals are available
for a given day. A detection of particular events is also performed in order to flag the data.
The processing of the data separates morning and afternoon orbits. The aggregated products
are generated from this fundamental product. The Level 4 SM, SMOS-L4 2.0 data (SMOS-



L4$^{2.0}$), with 1 km spatial resolution is provided by BEC and covers the IP, Balearic Islands,
Portugal, South of France, and North of Morocco (latitudes 34°– 45° N and longitudes 10° W
– 5° E). A downscaling method that combines highly accurate, but low-resolution SMOS
radiometric information with high-resolution, but low sensitivity, visible-to-infrared imagery
to SSM across spatial scales is used to derive the SMOS-L4$^{2.0}$ data (Piles et al 2010). The
impact of using different vegetation indices from MODIS with higher spatial and temporal
resolution in the downscaling method was explored in Sanchez-Ruiz et al. (2014), showing
that the use of more frequent and higher spatial-resolution vegetation information lead to
improved SM estimates. The latest SMOS-L4 product is the version 3.0 or "all weather"
(SMOS-L4$^{3.0}$), which is the product used and examined in this study. The downscaling
approach is based on Piles et al. (2014) and Sanchez-Ruiz et al. (2014), with the novelty of
introducing ERA-Interim Land Surface Temperature (LST) data in the MODIS LST/NDVI
scape. The evaluation of the SMOS-L4 2.0 and 3.0 products support the use of the "all
weather" version, since it does not depend on cloud cover and the accuracy of the estimates
with respect to in-situ data is improved or preserved (Piles et al. 2015 (Quality report)).
In this study, the SMOS-L2 V5.51 data coming from a L1C input product (obtained from
MIRAS measurements), the SMOS-L3 V2.72 and the SMOS-L4 V3.0 are employed.
2.3 The SURFEX-ISBA SVAT model
The SVAT model SURFEX (Externalized Surface, Le Moigne et al. 2009) – module ISBA
(Interactions between Soil-Biosphere-Atmosphere, Noilhan and Planton 1989) is used to
generate point-scale and spatially distributed SM spatial and temporal fields from initial
conditions and atmospheric forcing. SURFEX-ISBA was developed at the National Center for
Metorological Research (CNRM), at Météo France, and it has been widely validated over
vegetated and bare surfaces (e.g. Calvet et al. 1998). The ISBA scheme uses the Clapp and
Hornberger (1978) soil water model and Darcy's law for the estimation of the diffusion of





water in the soil, and allows 12 land use and related vegetation parameterization types. Crops
are considered for the VAS area since mainly vineyards, almond and olive trees and shrubs
compose the region.
The surface characteristics are considered in the SVAT input, roughness and the fraction of
vegetation are adopted from ECOCLIMAP (Masson et al. 2003), topography is obtained from
GTOPO (GTOPO30 Documentation) and soil types are defined using FAO (FAO, 2014).
To obtain an accurate simulation of soil moisture in the study area, the model was originally
calibrated by Juglea et al. (2010) to be applied over the entire site for any season/year.
Particularly relevant for this study is the specific definition of the soil hydraulic parameters
which they made for the VAS area, since most of the hydrological parameters are site
dependent. A new set of empirical equations as a function of the percentages of sand and clay
was defined using Cosby et al. (1984) and Boone et al. (1999). New definitions and
recommendations by Juglea et al. (2010) for the VAS area were adopted in this investigation.
*Atmospheric forcing information: ECMWF and SAFRAN*
High quality atmospheric forcing is needed to carry out accurate simulations. To run the
ISBA model, the following atmospheric forcing data are needed: air temperature and
humidity at screen level, atmospheric pressure, precipitation, wind speed and direction and
solar and atmospheric radiation. Three different sets of atmospheric forcing information are
used in this study; (a) meteorological data from 3 fully equipped stations in the OBS area,
MELBEX-I, MELBEX-II and VAS, (b) ECMWF (European Centre for Medium-Range
Weather Forecast) data, and (c) information from the SAFRAN (Système d'Analyse
Fournissant des Renseignements Atmosphériques à la Neige) meteorological analysis system
(Durand et al. 1999; Quintana-Seguí et al. 2008; Vidal et al. 2010).



Precipitation, air temperature, surface pressure, air specific humidity, wind speed and
direction, downward longwave radiation, diffuse shortwave radiation, downward direct
shortwave radiation, snowfall rate and $CO_2$ concentration are used as input data from the
meteorological stations aforementioned in the OBS area. A temporal resolution of 10 min is
available. From ECMWF, dew point and temperature at 2 m, pressure, precipitation and wind
components, are used as forcing data, with a 6 h temporal resolution and 0.125°x0.125°
spatial resolution. Precipitation, air temperature, surface pressure, air specific humidity, wind
speed and downward shortwave and longwave radiation from SAFRAN are used as input
information with a spatial resolution of 8x8 km$^2$ and an hourly temporal resolution. In this last
case, we have an optimal spatial and temporal distribution of the atmospheric forcing over the
VAS area (~ 50x50 km$^2$) and a rare to find complete database to force the land surface model.
More details about the SAFRAN system and its validation in north-eastern Spain could be
found in Quintana-Seguí et al. (2016).

## 3. Analysis methodology

In order to investigate the characteristics and potential added values of fine-scale SMOS-
derived soil moisture, the spatial variability, the temporal evolution as well as the probability
distribution is investigated. With this purpose, SMOS-derived soil moisture products at
different spatial resolutions, in situ measurements and model simulations are jointly
evaluated.
The spatial distribution and temporal evolution of precipitation and SMOS-derived soil
moisture over the IP and the VAS area are assessed for the time period from December 2011
to December 2012 considering also hydrological seasons (DJF: December-January-February,
MAM: March-April-May, JJA: June-July-August, SON: September-October-November).



During 2012, the Hydrological Cycle in the Mediterranean Experiment (HyMeX; Dobrinski et
al. 2014) took place in the Western Mediterranean with the IP and particularly the Valencia
area as target areas. During the SON period of 2012, the Special Observation Period (SOP1;
Ducrocq et al. 2014) with intensive experimental deployment over the area took place. This
provides us with valuable information about the environmental conditions as well as the
occurrence of precipitation events in the investigation area. SMOS-L3 (~ 25 km), SMOS-L2
(~ 15 km), and SMOS-L4$^{3.0}$ (~ 1km) are used for the evaluation of soil moisture distribution
at different grid spacing. Piles et al. (2014) pointed out that differences may exist between
SMOS-L3–L2 and the 1 km disaggregated soil moisture SMOS-L4 because of the distinct
methodology used to obtain these products. Only SMOS descending passes or a mean
between ascendant and descent passes are used to calculate mean daily values of SMOS-
derived soil moisture. Soil moisture derived from the afternoon orbits was found to be more
accurate than the morning passes (Piles et al. 2014). The fine temporal resolution of the model
simulations (1 h) and the observations (10 min) allow comparisons at the time of the SMOS
overpasses. Because of the 3-day revisit period of the SMOS swath, the IP will not be fully
covered by the satellite on daily basis. However, despite identified difficulties (radio
frequency interferences, missing data ...), the IP is well observed being 1.5 days the average
observations frequency over the IP. Only those images with coverage higher than 50% are
considered in our calculations. A conservative remapping to coarser resolutions is applied,
when required, to make comparisons among each other or with respect to ground-based
observations on equal terms. Remapping allows point to point comparisons between these
data sets. In addition to the yearly and seasonal approach, an exemplary short time period, 19
to 20 October of 2012, is considered. These correspond to the periods in which two extreme
precipitation events occurred, affecting south and eastern Spain (end of September; Khodayar
et al. (2015)) and the Ebro valley (at the end of October; Jansà et al. 2014), respectively.
Therefore, high variability in the soil moisture distribution is expected.





The coefficient of variation (CV), defined as the ratio of the standard deviation to the mean,
of the precipitation and soil moisture fields over the IP, the VAS (50x50 km2) and the OBS
(10x10 km2) area are examined for the analysis of the spatial variability of the
aforementioned fields. The soil moisture daily index ($SM_{index,i}$) is calculated to assess the
evolution pattern allowing the study of daily variations
$SM_{index,i} = (SM_{i+1} - SM_i)/ SM_i$, where $SM_{i+1}$ is the soil moisture of the day i+1 and $SM_i$ is the
soil moisture of the day before i.
The reliability of SMOS-L2 and SMOS-L4$^{3.0}$ soil moisture products is evaluated by
comparison with in situ soil moisture measurements in the OBS area. The spatial and
temporal variability are considered as well as the probability distribution. Different
approaches are applied: (a) the nearest grid point is selected for point-like comparisons
between SMOS-L2 and SMOS-L4$^{3.0}$ against in situ soil moisture stations, to reduce sampling
biases in this region of diverse soil characteristics (Table 1), (b) SMOS-L4$^{3.0}$ soil moisture
grid cells are averaged over the 10x10 km$^2$ area and compared to the mean from the soil
moisture network stations to address the issue related to spatial averaging. For the comparison
between the SMOS-L2 and the in situ observations: when single ground-based stations are
considered the closest SMOS pixel is selected, in case of considering the OBS (10x10 km$^2$) or
VAS (50x50 km$^2$) areas the mean over all pixels which centre falls within the area is used.
For the comparison with SMOS descending passes the corresponding values from in situ
measurements are considered. Additionally, a separation between wet days (precipitation over
1 mm/d) and dry days is applied to consider possible implications of wet/dry soils for SMOS
measurements.
Linear regression, the coefficient of determination ($R^2$), the mean bias (MB), and the root
mean square deviation (RMSD) are used to predefine the accuracy. A debiased or centred



RMSD (CRMSD) is applied to discriminate the systematic and random error components
removing the overall bias before calculating the RMSD.
Soil moisture modeling is performed by the use of the SVAT, SURFEX (Externalized
Surface) – module ISBA (Interactions between Soil-Biosphere-Atmosphere) from Météo-
France. Configuration and specifications described in Juglea et al. (2010), which proved
successful in adequately simulate the associated soil moisture heterogeneity over the wide
VAS surface (50x50 km$^2$), are adapted in this study. Simulations start on 1 December 2011 at
00UTC and cover the whole investigation period until 31 December 2012 with an hourly-
output time resolution. Point-scale SURFEX-ISBA simulations over the soil moisture
network stations in the VAS domain are validated with the in situ measurements to assess the
usefulness of the model for further investigation, picturing the potential of the model in
simulating upper level soil moisture variability on different soil characteristics (Table 1). The
impact of different soil moisture initializations on the temporal evolution of upper-level soil
moisture is additionally evaluated using initialization perturbation simulations. Since
measurements in the area are available since 2003, a climatological mean is calculated for
each of the soil moisture stations and considered for initialization of the control simulations
(CTRL). Three additional initialization experiments are performed, a) with the daily mean of
the real observation (ground-based measurement) on the initialization day, b) the
climatological seasonal mean, c) the climatological monthly mean.
To try to simulate the spatial and temporal heterogeneity of the soil moisture fields over the
VAS surface, the SURFEX-ISBA scheme is used in combination with high quality forcing
data from ECMWF (hereafter SURFEX-ECMWF) and the SAFRAN system (hereafter
SURFEX-SAFRAN) for spatialization purposes. The benefit of initializing the simulations
with SMOS-L4$^{3.0}$ data in comparison to climatological means is discussed. Two exemplary
initializations - in a wet period and a dry period are examined. A comparison between



SURFEX-SAFRAN point-scale and 10x10 km$^2$ mean simulations is done against ground
measurements to assess the accuracy of the simulated SSM maps.

**4. Results**
4.1 SMOS-derived soil moisture at different resolutions
4.1.1 Spatial variability on seasonal and sub-seasonal time scales
Atmospheric forcing, evapotranspiration (ET), soil texture, topographical features and
vegetation types have been recognized as relevant factors contributing to soil moisture
variability (Rosenbum et al. 2012). The response of soil moisture to precipitation changes
largely depends on soils water capacity and climatic zones. Particularly, in dry climates such
as the IP, soil moisture quickly reacts to changes in precipitation (Li and Rodell 2013).
Precipitation variability and mean are positively correlated, thus, an increase in precipitation
yields wetter soils, which in turn results in higher spatial variability of soil moisture.
In the autumn period, the western Mediterranean is characterized by a large thermal gradient
between the atmosphere and the sea (Duffourg and Ducrocq, 2011, 2013) resulting in intense
precipitation extremes (Raveh-Rubin and Wernli 2015). Precipitation in the IP during the
autumn (SON) period of 2012 was above average (Khodayar et al. 2015). It is also the
hydrological season in which higher variability in the soil moisture is observed as a result of
the precipitation distribution (period used hereafter for investigation). The positive anomaly is
largely caused by two unique events, i.e. at the end of September (27-29) affecting south and
eastern Spain and at the end of October (19-20) affecting the Ebro valley (Jansà et al. 2014).
Figure 2a shows the north-south precipitation gradient for the SON period mean. The SSM
satisfactorily reflects this gradient (Figure 2b), but, more markedly for the SMOS-L3 and



SMOS-L2 than the higher resolution SMOS-L4$^{3.0}$ showing lower standard deviation, SMOS-
L3(~0.15±0.01), SMOS-L2(~0.17±0.01), SMOS-L4(~0.22±0.007). The same performance is
seen over the VAS domain (not shown). The SSM variability associated to the extreme
precipitation events in this period is not well represented in the SMOS-L4$^{3.0}$ seasonal mean.
Table 2 shows the number of days (percentage) in which there is more than 50 % of data over
the IP for each SMOS product. These periods have been used as basis for the calculation of
the spatial distributions in Figure 2b. SMOS-L3 (88 %) and SMOS-L2 (84 %) show a good
coverage and similar number of days. However, a large difference is observed with respect to
the SMOS-L4$^{2.0}$ product with only 28 days (32 %) of adequate coverage for the period of
SON 2012. This is due to the problematic associated to the downscaling approach used to
obtain the 1 km soil moisture maps, in which the lack of Land Surface Temperature (LST)
information from MODIS visible/infrared (VIS/IR) satellite data in cloudy conditions (section
2.2) constrains derived-SSM information. The availability and usefulness of this product is
therefore significantly reduced. The new product L4$^{3.0}$, used in this study, in which the
previous limitation is resolved using ERA-Interim-derived LST information, shows a
coverage percentage in the order of 92 %, even higher than the SMOS-L3 and -L2 products.
However, Figure 2b demonstrates that the spatial representation of the seasonal mean does not
improve with this product, as a consequence of the limited temporal availability of the
SMOS-derived SSM product dictated by the revisit period of the satellite.
In Figure 3, only common available days from all different operational levels are selected for
an inter-SMOS product comparison. When remapped to the same resolution (coarser grid
spacing) comparable values are identified between SMOS-L3, -L2 and –L4$^{3.0}$ for the JJA and
SON period, whereas relevant differences are pointed out from December to May. In this last
period, we identify higher means for the SMOS-L4$^{3.0}$ product and SMOS-L3 with respect to
SMOS-L2, which is in agreement with a systematic dry bias identified for SMOS-L2 also in
previous studies (section 1).
At sub-seasonal scales, e.g. event scale on the 19-20 November 2012 (Figure 4), the SMOS-
L4$^{3.0}$ product shows SSM mean and variability in the same range of the SMOS-L2 and -L3
products, but with a finer-improved resolution representation of the spatial distribution.
Comparisons with the mean ground-based SSM at the VAS (OBS area: *0.25 ±0.0002*) show
better agreement with the mean SSM from the SMOS-L4$^{3.0}$-1 km disaggregated product
(*0.23±0.002*) and poorer correlation with SMOS-L2 (*0.20±0.002*). The problematic of SMOS-
L4$^{3.0}$ on seasonal time scales vanishes at sub-seasonal (event) scales where the potential
added value of the 1 km product is manifest.
4.1.2 Temporal evolution of surface soil moisture data sets
The SMOS and in situ measured SSM time series are investigated in this section. To assess
the behaviour and variability of these data sets we consider, (a) the *soil moisture daily index,*
to investigate the pattern of such evolution based on daily variations, and (b) the *coefficient of*
*variation (CV)*, for the analysis of the spatial variability and its evolution in time (Figure 5).
The temporal behavior of SSM averaged over the IP, the VAS domain, and the OBS area are
compared in Figure 6. SMOS afternoon (descendant; Piles et al. 2014) orbits are selected as
well as observations at the time of the SMOS overpasses. For the IP and VAS, SMOS-L2 and
SMOS–L4$^{3.0}$ have been remapped to the coarser grid spacing for an adequate comparison.
Ground-based observations are aggregated using a mean over all stations for comparison with
the corresponding SMOS-L4$^{3.0}$ data (the closest grid point is selected).
Overall, the averaged SMOS-L2 and -L4$^{3.0}$ data over the IP are much more variable than the
SMOS-L3, showing a more extreme daily index (SMOS-L2: -1 to 2; SMOS-L4$^{3.0}$: -0.7 to
1.45). Over the VAS, SMOS-L2 is clearly more variable than the higher resolution SMOS-



L4$^{3.0}$. But, the last one shows a wider range of values as well as more extreme daily index
values when compared to the averaged in situ soil moisture measurements. The CVs of the
spatially averaged SMOS-L4$^{3.0}$ is lower than those of SMOS-L3, -L2 and in situ observations
indicating that this data are less scattered. Despite detected differences within in situ
observations, SMOS responds well to soil moisture variations over time.
Although absolute values are not totally captured, all three SMOS products adequately
reproduce the temporal dynamics at a regional scale. The systematic dry bias present on
SMOS-L2 data (Piles et al. 2014) is evident particularly on the first half of the year. A mean
bias in the order of -0.09 to -0.07 m$^3$/m$^3$ is identified for the DJF-MAM period; this difference
is reduced to -0.02 m$^3$/m$^3$ for the JJA-SON period (Table 3). During the DJF-MAM period the
vineyards are bare, only the vine stocks are present. The water content of the vine stocks
negatively impacts the SMOS measurements (Schwank et al. 2012).
Good agreement is found between the SMOS-L4$^{3.0}$ product and the mean of the in situ
observations (the network's variability (shaded grey) contains the SMOS-L4$^{3.0}$ data). Scores
confirm this result particularly for the periods DJF and SON (slope~1, R$^2$~0.7). Poorer
correlation is found for the MAM (slope~0.6, R$^2$~0.4). In this period, soil moisture maxima
immediately after the precipitation events are not always well captured by the SMOS-L4$^{3.0}$
data, showing additionally a too rapid drying after this. This observation agrees with the
SMOS' inability of correctly measuring in situations when liquid water is present at the soil.
The measured signal is perturbed during the vegetation growing season, which could explain
the worse statistics. On the other hand, during JJA, low slope~0.1 and R$^2$~0.01 could be in
relation to SSM values close to or lower than 0.1 m$^3$/m$^3$ and very low spatial variability,
which was found to be necessary for an adequate performance of the algorithm used for the
derivation of the SMOS-L4 1 km product in Molero et al. (2016).
4.2 Spatial comparison at high-resolution: SMOS-L4$^{3.0}$ versus ground measurements




High-resolution spatio-temporal correlations are assessed by spatial comparison with in situ
observations. Characteristics of each of the in-situ stations are presented in Table 1. A
seasonal analysis is performed focusing on the selected year of measurements covering a
complete hydrological cycle (from 1 December 2011 to 31 December 2012). Comparisons
between SMOS-L2 and ground measurements are additionally included.  In Figure 7, the
scatter plots display (a) possible differences between dry and wet days (> 1 mm/d), and (b, c)
the agreement between remotely sensed and in situ soil moisture measurements from the OBS
network using the seasonal classification. To consider any uncertainties arising from spatial
averaging, ground measurements are compared to point like and 10x10 km$^2$ SSM means. The
10x10 km$^2$ area used covers the OBS area, i.e., the network of in situ measurements within
the VAS. For comparison, all grid points from SMOS-L4$^{3.0}$ and SMOS-L2 included within
the area are considered. Statistics for individual comparisons at all stations are summarized in
Table 3.
In Figure 7a, the separation between days with and without precipitation (< 1 mm/d) points
out similar correlations during dry than wet days (RMSD~0.015, R$^2$~0.7) for SMOS-L4$^{3.0}$,
whereas a slightly better agreement is found for the dry days (not shown) for SMOS-L2. A
systematic mean dry bias of about 0.05 (dry days) to 0.08 (wet days) m$^3$/m$^3$ is assessed for
SMOS-L2, while a lower bias with changing sign is identified for the L4$^{3.0}$ product (~ 0.005
(wet days); ~ -0.02 (dry days)). Comparisons using the corresponding mean over the 10x10
km$^2$ OBS area, in Figure 7b and Table 3, show good agreement with respect to the SMOS-
L4$^{3.0}$ and poorer scores for SMOS-L2 (only one grid point of SMOS-L2 is located within the
OBS area). Worse consistency is found in both cases for the MAM and JJA periods. CRMSD
is in all cases in the required range of ≤ 0.04 m$^3$/m$^3$. Point-like comparisons with the
individual in situ stations, in Figure 7c and Table 3, show that spatial patterns are captured at
1km with RMSD~0.007 to 0.1 m$^3$/m$^3$ but, in most cases, accuracy for SMOS-L4$^{3.0}$-1 km





disaggregated product is within the required range of less than 0.04 m$^3$/m$^3$ (not shown).
Higher RMSD is found for SMOS-L2, ~ 0.008 to 0.13 m$^3$/m$^3$, accounting for the previously
identified dry bias (~ (-0.14) – (-0.02)) reduced in SMOS-L4$^{3.0}$ (~ (-0.08) – (-0.01)). The
CRMSD is in all cases ≤ 0.04 m$^3$/m$^3$. For all stations, better correlations are found in DJF and
SON and poorer scores in JJA and MAM, in agreement with the areal-mean comparisons
(section 4.1.3). Best scores are obtained for Nicolas, VAS and La Cubera stations, probably in
relation to their common soil type distribution, over vineyards, and homogeneous conditions,
over a plain (Figure 8a, Table 3). The SON time period reveals the best agreement, at this
time the vineyards are completely grown (however, senescent thus containing less water) and
SSM exhibits substantial spatial variability driven by precipitation and irrigation thus
improving spatio-temporal correlations. Worse statistics are found for Melbex-I, Melbex-II
and Ezpeleta, probably in relation to the location of the soil moisture probes in rockier and
orographically more complex areas, also in proximity to forestall and man-made construction
areas.
The soil moisture probability distribution function (PDF; Figure 8b) of all in situ
measurements versus SMOS-L4$^{3.0}$ data reveals that the later overestimates SSM below 0.1
m$^3$/m$^3$, values mainly observed during the JJA period. But, an underestimation occurs in the
range between 0.1 and 0.3 m$^3$/m$^3$, which is consistent with the identified underestimation of
maximum soil moisture reached after a precipitation event and the rapid drying of the soil in
comparison to the much slower response seen in the observations during the MAM period
(Figure 6c).
4.3 SURFEX model simulations and realistic initialization with 1-km soil moisture data
4.3.1 SURFEX model simulations of selected stations and realistic initialization



Land surface models are commonly used to analyse regional soil moisture estimates.
Initialization of land surface models is a crucial issue and its impact on the accuracy of model
estimation is widely recognized to be significant. When observations are not available, soil
moisture initialization is generally performed with simulated climatological mean values. In
this section, different sensitivity experiments with the SURFEX-ISBA SVAT model are
performed to investigate the impact of initialization in the simulation of the spatio-temporal
evolution of point-scale soil moisture and regional SSM fields.
As a first step, the performance of the SURFEX-ISBA SVAT model is evaluated. SURFEX-
ISBA point-like simulations are performed for all in situ soil moisture stations at the VAS
area to assess the usefulness of the model for further investigation (Table 4). To obtain an
accurate simulation of soil moisture in the area, the model has been calibrated and particular
characteristics have been considered following the recommendations by Juglea et al. (2010)
for each of the stations. The complete hydrological cycle (from 1 December 2011 to 31
December 2012) is simulated for each station.
SURFEX-ISBA simulations show good agreement with soil moisture ground-based
observations at all stations, adequately capturing the associated spatio-temporal variability
(slope~1, $R^2$~ 0.7 to 0.9; MB~0.1 $m^3/m^3$; CRMSD~0.02 $m^3/m^3$). It can be concluded that the
model performs well and is therefore suitable for further investigation. The seasonal analysis
points out the best simulations in the SON period ($R^2$~0.9 for all stations), but CRMSD is $\leq$
0.04 $m^3/m^3$ for all stations at all periods.
Four experiments are performed modifying the initial soil moisture scenario using: (a) the
mean of the ground-based measurement on the day of the initialization (realistic initialization;
REAL-I), (b) the mean over the December month from the ground-based measurements
(MONTH_I), (c) the seasonal mean (DJF) from the ground-based measurements (SEASON-I)
and (d) the climatological ground measurements soil moisture mean over the last 10 years for



the December period (Figure 9a). Deviations of the sensitivity experiments with respect to the
mean of ground measurements reveal an impact during the whole simulation period even
though initial scenarios were close to each other. Even after strong precipitation events, which
reduce RMSD, the soil moisture evolution is affected by the initialization. REAL-I
simulations show the best agreement with in situ observations ($R^2 \sim$ 0.9; CRMSD~ 0.02
$m^3/m^3$). Thus, this realistic initial scenario based on in situ soil moisture observations is
hereafter used for model initialization in our control experiments. Temporal mean
comparisons for each station are presented in Figure 9b and Table 4 using the above described
REAL-I initialization scenario.
4.3.2 Spatialization
With the purpose of simulating soil moisture over a whole SMOS pixel, Juglea et al. (2010)
combined the SURFEX-ISBA model and ground and meteorological observations in the
study area. In this section, to obtain an accurate mapping of soil moisture over the VAS
(50x50 $km^2$) we discuss a different methodology for the spatialization of SURFEX-ISBA
simulations. Atmospheric forcing information both from ECMWF and SAFRAN is used as
input data (hereafter SURFEX-ECMWF and SURFEX-SAFRAN simulations, respectively).
SMOS-L$4^{3.0}$-1 km disaggregated values are used for initialization. In-situ soil moisture
observations over the VAS area are considered for verification. Soil moisture initialization in
spatialized SURFEX simulations requires a single representative value for the whole
simulation area. In this case, we use as input the SMOS-L$4^{3.0}$-1 km disaggregated soil
moisture mean over the whole simulation area for the initialization day. For comparison, the
mean of all ground-based observations is also used for the initialization.
As a first step, point-scale SURFEX-ECMWF and SURFEX-SAFRAN simulations covering
the whole investigation period are performed for all in situ soil moisture stations to examine
its ability to reproduce soil moisture dynamics. Ground measurements at each station are used



for initialization. Scores clearly indicate better agreement with all in situ observations for the
SURFEX-SAFRAN simulations (slopes~ 1, $R^2$~ 0.9, RMSD< 0.1 m$^3$/m$^3$), rather than the
SURFEX-ECMWF simulations (slopes> 1, $R^2$~ 0.6, and RMSD> 0.1 m$^3$/m$^3$).
In a second step, SURFEX-ECMWF and SURFEX-SAFRAN simulations are spatialized to
obtain maps of soil moisture over the investigation area. In our CTRL simulations, the daily
soil moisture from the mean of the in-situ measurements on the initialization day is used for
model initialization. Mean SSM from in situ measurements for the whole investigation period
is in the order of 0.14±0.005, whereas SURFEX-ECMWF derived SSM field is about
0.18±0.007 and SURFEX-SAFRAN derived SSM field is 0.15±0.002, thus, closer to ground-
based observations. Performing a seasonal analysis, we demonstrate that this consistency is
maintained for all seasons (not shown). The higher resolution of the SAFRAN-atmospheric
forcing better reproduces the high spatial heterogeneity over the VAS area resulting in
improved mapping of simulated SSM.
Initialization of the SURFEX-SAFRAN combination using SMOS-L4$^{3.0}$ is examined. Two
sensitivity simulations are performed using for the initial soil moisture scenario, (a) the daily
soil moisture mean from the SMOS-L4$^{3.0}$ data (which is generally close to observations; EXP-
SMOS), and (b) the climatological soil moisture from observations (daily mean over 10 years,
which has been selected to be far from observations; EXP-CLIM). These experiments are
initialized in dry periods, following Khodayar et al. (2014) recommendations, to maximize
the impact, and run for about 3-4 months. In the first case, initialization is performed in a
winter month (December) and the whole simulation period remains almost dry. In the second
case, a summer month (July) is chosen for the initialization and it is followed by a wet autumn
period with frequent heavy precipitation events in the area.
The temporal evolution of the RMSD (Figure 10a) demonstrates that the initial soil moisture
scenario influences its evolution until the end of the simulation, in agreement with previous





results in section 4.3.1. Larger deviations occur during dry periods, in both scenarios. Longer
spin-up times, defined as the time that soil needs to reestablish quasi-equilibrium, characterize
the dry scenario. It is after heavy precipitation events that deviations decrease. Soil quickly
reacts to changes in the precipitation field in the semi-arid IP. When the upper level soil gets
close to saturation soil memory is almost lost. Before the high precipitation events, SSM
evolves following the direction of the initial perturbation, i.e., higher initial SSM yields
higher SSM, however, a stochastic behaviour is identified afterwards.
As an example, differences in the spatial distribution of soil moisture for the winter/dry period
simulation are discussed (Figure 10b). A relevant difference in the mean is identified when
compared to the CTRL simulation (0.17±0.004): EXP-CLIM (0.014±0.003), EXP_SMOS
(0.17±0.003). Clearly, better agreement is found in this last case.
Considering the EXP-SMOS initialization scenario simulation, a comparison between
simulated point-like and the 10x10 $km^2$ mean against corresponding ground measurements
was done for verification (Figure 10c). Correlations in the order of $R^2$~0.9 confirm that the
combined use of SURFEX-SAFRAN and SMOS-L4$^{3.0}$ for initialization successfully
reproduces soil moisture spatial and temporal variability becoming an optimal tool for
mapping soil moisture heterogeneity over a study region for diverse purposes.

## 580    5. Discussion and conclusions

High-resolution soil moisture products are essential for our understanding of hydrological and
climatic processes as well as improvement of model skills. Due to its high spatial and
temporal variability, it is a complicated variable to assess. Mapping high-resolution soil
moisture fields using intensively collected in-situ measurements is infeasible. Thus, state of
the art high-resolution modelling and satellite-derived products have to fill this gap, although





verification is needed. In this study, we provide information about the advantages and
drawbacks of soil moisture SMOS satellite products at different operational levels examining
the potential of the state of the art SMOS-L4$^{3.0}$-1 km disaggregated product for assessment of
soil moisture variability, and improvement of SVAT simulations through realistic model
initialization. The proposed analysis focuses on the semi-arid IP and covers the one year
period of 2012 (from December 2011 to December 2012).
The SMOS-L4$^{3.0}$-1km product is compared to different resolution soil moisture data products
from SMOS, namely SMOS-L3 (~ 25 km) and SMOS-L2 (~15 km). Their ability in
reproducing soil moisture dynamics and heterogeneity and the added value of SMOS-L4 is
examined using a dense network of ground-based soil moisture measurements over the
Valencia Anchor Station (VAS; one of the SMOS test sites in Europe) for verification.
Perturbation simulations of point-scale surface soil moisture are investigated to assess the
sensitivity to soil moisture initialization. The Soil Vegetation Atmosphere Transfer (SVAT)
model SURFEX (Externalized Surface) – module ISBA (Interactions between Soil-
Biosphere-Atmosphere) is employed. Furthermore, the SURFEX-ISBA model is used in
combination with the ECMWF forcing information (SURFEX-ECMWF) and the SAFRAN
meteorological analysis system (SURFEX-SAFRAN) to obtain a representative soil moisture
representation over the VAS area. The sensitivity of the SURFEX-SAFRAN scheme to
simulate the heterogeneity of surface soil moisture applying realistic initialization with
SMOS_L4$^{3.0}$ ~1 km product is investigated.
Correlation with precipitation is traceable in the temporal evolution of in situ ground
measurements and SMOS-derived soil moisture products. On seasonal time scales, SMOS-L3
(~ 25 km) and SMOS-L2 (~15 km) adequately represent the soil moisture gradient and high
soil moisture episodes in relation to the precipitation distribution. However, the seasonal
representation of SMOS-L4$^{3.0}$-1 km soil moisture does not capture these maxima despite the





novelty of introducing ERA-Interim LST data in the MODIS LST/NDVI space (Piles et al. 2014; Sanchez-Ruiz et al. 2014), probably due to the so different spatial resolution of ERA-Interim and MODIS. This new downscaling approach greatly enhances the potential applicability of the data for those days/periods in which measurements are available, but cannot fill in those periods without measurements dictated by the revisit period of the SMOS satellite, hence, compromising the soil moisture representation as a mean for longer periods than a day. On sub-seasonal time scales, when SMOS images are available, the SMOS-L4$^{3.0}$ high-resolution product shows its potential. It adequately captures the surface soil moisture variability in association with the precipitation field, also when extreme precipitation takes place.

Characteristics of SMOS-L4$^{3.0}$ soil moisture fields are closer to in-situ observations than SMOS-L3 and -L2 products. Comparisons with in-situ measurements reveal that, generally, all three SMOS products adequately reproduce the soil moisture temporal dynamics meeting the desired accuracy of the mission (0.04 m3/m3); however, the spatial patterns did not reach the expected precision in agreement with former studies in other regions (Gonzalez-Zamora et al. 2015). The contrast between point-scale in-situ measurements and the coarse resolution of the satellite observations is an issue that should be considered. A systematic dry bias, particularly evident in the first half of the year (December to May), is identified in the SMOS-L2 data, also observed in former investigations. The negative impact of the water content of the vine stocks (vineyards are bare in this time period) on the SMOS measurements and the coarser resolution result in poorer scores of the SMOS-L2 when compared to in-situ observations. The SMOS-L4$^{3.0}$ product and the mean of the in-situ observations show a good agreement in general. This is consistent with the finer resolution of this product which better captures local information on the 1 x 1 km pixel, whereas coarser products smooth out this vital information.





The SMOS-L4$^{3.0}$ soil moisture probability distribution function (PDF) in comparison to that
of the in-situ measurements reveals a SMOS overestimation below 0.1 m$^3$/m$^3$ and an
underestimation in the range between 0.1 to 0.3 m$^3$/m$^3$. A seasonal analysis points out better
scores for the DJF and SON periods, whereas poorer correlation is found for the MAM and
JJA periods. In the MAM period, an under-representation of the rainy events is found, as well
as faster and stronger drying changes coinciding with the vegetation growth season. In JJA,
the very low soil moisture values (< 0.1 m$^3$/m$^3$) with associated low spatial variability results
in low R$^2$. During dry and wet days (> 0.1 mm/d), similar correlations are found for SMOS-
L4$^{3.0}$ comparisons with in-situ observations. A low bias with changing sign is identified for
the L4$^{3.0}$ product (~ 0.005 (wet days); ~ -0.02 (dry days)). SMOS-L2 reveals slightly better
agreement for the dry days and a systematic mean dry bias of about 0.05 (dry days) to 0.08
(wet days) m$^3$ /m$^3$.
Point-like and 10x10 km$^2$ comparisons show good agreement with respect to the SMOS-L4$^{3.0}$
and poorer scores for SMOS-L2 (e.g. DJF period: SMOS-L3/-L2: Slope:1.1/1.0, R$^2$:0.5/0.7,
Bias:-0.09/(-0.03)). CRMSD is in the required range of ≤ 0.04 m$^3$/m$^3$ in most cases.
Comparison of the SMOS-L4$^{3.0}$ data with ground soil moisture measurements from the eight
stations in the network (10x10 km$^2$) over the VAS area shows that the spatial patterns are
captured at 1 km with RMSD~ 0.007 to 0.1 m$^3$/m$^3$ (5 out of the 6 stations investigated show
an accuracy of less than 0.04 m$^3$/m$^3$, benchmark of the SMOS mission). The best correlations
are in DJF and SON, and poorer scores in MAM and JJA, in agreement with the areal-mean
comparisons. SMOS-L4$^{3.0}$ data shows better agreement at those stations over plain areas and
with uniform conditions (vineyards), against those over more complex and less homogeneous
terrains (rocky soils and areas close to forestall and man-made constructions).
The impact of initialization scenarios on the simulation of SSM is investigated by means of
SURFEX-ISBA SVAT simulations. Firstly, the performance of the land surface model is





evaluated. Simulations covering the whole investigation period over all in-situ measurement stations at the VAS area have been carried out. In all cases, simulations show good agreement with ground-based observations. Mean values are well reproduced for all stations and the temporal variability is well captured (R2~0.7 to 0.95; RMSD~0.02). Four sensitivity experiments using different initial scenarios are performed, (a) the mean of the ground-based measurement on the day of the initialization (realistic initialization; REAL-I), (b) the mean over the December month from the ground-based measurements (MONTH_I), (c) the seasonal mean (DJF) from the ground-based measurements (SEASON-I) and (d) the climatological soil moisture mean over the last 10 years for the December period. Deviations larger than zero are present during the whole simulation period demonstrating the impact of the initial soil moisture scenarios on its temporal evolution, even when close initial conditions are considered. As expected, the use of real observations on the initialization day shows the best agreement ($R^2 \sim 0.9$; CRMSD$\sim 0.02$ m$^3$/m$^3$).

In a further step, SURFEX-ISBA simulations are combined with ECMWF and SAFRAN atmospheric forcing information to obtain soil moisture maps over the VAS domain. The higher resolution of the SAFRAN forcing data as well as the larger number of input variables result in higher correlations with in-situ SSM measurements, hence, offering a good base for investigating the potential impact of the soil initialization with SMOS-L4$^{3.0}$-1 km disaggregated soil moisture.

The sensitivity of SURFEX-SAFRAN SSM field simulations to an initialization with realistic SSM values from the SMOS-L4$^{3.0}$ data set is compared to that using daily climatological means. The model is initialized in a winter month (December) and in a summer month (July) and runs free from this point to about 3-4 months, covering a dry and a wet period, respectively. It may be concluded that in both cases, positive differences are present until the end of the simulations. The largest deviations are found during dry periods in both scenarios.



Soil is more sensitive to initialization during dry periods, i.e., longer spin-up times (time the
soil needs to restore quasi-equilibrium) are needed. RMSD is in both periods closer to zero
after heavy precipitation events. The upper level soil moisture rapidly reacts to precipitation,
soil conditions close to saturation result in the loss of soil moisture memory in the upper soil
level. The long-term impact of the initial dry or wet scenario, acts in a stochastic way after
heavy precipitation events, independently from the sign of the initial perturbation. Good
agreement was reached when comparisons between point-like and 10x10 $km^2$ simulations
with SURFEX-SAFRAN initialized with SMOS-L4$^{3.0}$ data and in-situ soil moisture
measurements were made ($R^2$~0.9 and RMSD<0.04 $m^3/m^3$).
In this study, the comparison and suitability of different operational satellite products from the
SMOS platform is investigated to provide realistic information on the water content of the
soil. The comparison carried out helps drawing guidelines on best practices for the sensible
use of these products. Currently, there is not a consensus about what is the "best" SMOS
product. Different users utilize different products depending on their application rather than
based on performance arguments. This study and the conclusions obtained on the comparison
are important to provide information on the advantages and drawbacks of these datasets.  The
high temporal and spatial resolution soil moisture maps obtained in this study could be of use
to build climatologies of SSM, as initial condition for convective system modelling, for flood
forescasting and for downstream local applications such as crop monitoring and crop
development strategies. Additionally, an accurate representation of SSM will permit the
calculation of SM profiles by application of e.g. exponential filters, which has been
demonstrated to be a successful technique. This is however, out of the scope of the paper, and
will be investigated in a follow-up research activity. Furthermore, the added value of the
SMOS-L4$^{3.0}$-1 km disaggregated product for initialization purposes is demonstrated, which
suggests its potential for assimilation purposes. Nevertheless, important aspects of the SMOS-



L4$^{3.0}$ SSM product have still to be improved, namely its temporal availability (e.g. successful
investigations on the increase of SMOS-L3 temporal resolution to 3h are available (Louvet et
al. 2015)), its spatio-temporal correlation with in situ measurements over complex
topographic areas, in areas/periods with low spatial variability and in rainy periods when an
under-representation and rapid decay of SSM has been identified.

















**Acknowledgements**

The authors acknowledge AEMET for supplying the precipitation data and the HyMeX

database teams (ESPRI/IPSL and SEDOO/Observatoire Midi-Pyrénées) for their help in

accessing the data. The SMOS products were obtained from CATDS (Centre Aval de

Traitement des Données SMOS) and SMOS-BEC (Barcelona Expert Center. We

acknowledge the support of the SURFEX-web team members. The ECMWF data was

obtained from http://www.ecmwf.int. Special thanks go to Pere Quintana for providing the

SAFRAN atmospheric forcing data. A. Coll´s work was supported by both National Spanish

Space Research Programme projects MIDAS-6 (MIDAS-6/UVEG. SMOS Ocean Salinity and

Soil Moisture Products. Improvements and Applications Demonstration) and MIDAS-7

(MIDAS-7/UVEG. SMOS and Future Missions Advanced Products and Applications). The

first author's research is supported by the Bundesministerium für Bildung und Forschung

(BMBF; German Federal Ministry of Education and Research).





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





**Tables**

**Table 1:** Characteristics of soil moisture stations within the VAS domain.

| NAME | STATION | DOMINANT VEGETATION USED FOR SIMULATIONS | TYPE OF VEGETATION | SAND | SILT | CLAY | ALTITUDE (m) | ANNUAL MEAN TEMPERATURE (°C) | ANNUAL MEAN PRECIPITATION (mm) |
|---|---|---|---|---|---|---|---|---|---|
| Melbex_I | | Schrub | Schrub | 0,47 | 0,38 | 0,15 | 849 | | |
| Nicolas | | Vineyard | Schrub/ Vineyard | 0,47 | 0,35 | 0,18 | 859 | | |
| La Cubera | | Vineyard | Vineyard | 0,45 | 0,35 | 0,20 | 762 | (12-14) | 451 |
| Ezpeleta | | Olive tree | Olive tree | 0,44 | 0,39 | 0,17 | 781 | | |
| VAS | | Vineyard | Vineyard | 0,46 | 0,37 | 0,17 | 804 | | |
| Melbex_II | | Vineyard | Vine stump/ Vine row | 0,45 | 0,29 | 0,26 | 797 | | |






















**Table 2:** Number of days (percentage) in which the SMOS (ascendant and descendent
swaths) coverage is higher than 50 %.

| LEVEL SMOS | SEPTEMBER | | OCTOBER | | NOVEMBER | | SON | |
|---|---|---|---|---|---|---|---|---|
| | days | % | days | % | days | % | days | % |
| L4$^{2.0}$ (~1km) | 10 | 34 | 9 | 31 | 9 | 31 | 28 | 32 |
| L4$^{3.0}$ (~1km) | 23 | 74 | 29 | 90 | 30 | 100 | 82 | 92 |
| L2 (~15km) | 20 | 67 | 28 | 90 | 28 | 93 | 76 | 83 |
| L3 (~25km) | 22 | 73 | 29 | 93 | 29 | 96 | 80 | 88 |

























**Table 3:** Statistics of daily areal averages of SMOS-L2 and SMOS-L4[3.0] soil moisture versus
ground-based soil moisture measurements over OBS. SMOS descendent orbits are selected
for the comparison.


| OBS vs SMOS-L2 | Slope | R2 | Bias | CRMS | OBS vs SMOS-L4[3.0] | Slope | R2 | Bias | CRMS |
|---|---|---|---|---|---|---|---|---|---|
| DJF | 1.1 | 0.5 | -0.09 | 0.03 | DJF | 1.0 | 0.7 | -0.03 | 0.04 |
| MAM | 0.6 | 0.2 | -0.07 | 0.03 | MAM | 0.6 | 0.4 | -0.03 | 0.03 |
| JJA | 0.3 | 0.01 | -0.02 | 0.03 | JJA | 0.1 | 0.01 | -0.003 | 0.03 |
| SON | 1.1 | 0.8 | -0.02 | 0.04 | SON | 0.8 | 0.7 | -0.003 | 0.04 |


| SMOSL2 vs SMOSL4[3.0] | M-I | M-II | VAS | NIC | EZ | LC | OBS (mean all stations) |
|---|---|---|---|---|---|---|---|
| **DJF** | | | | | | | |
| Slope | 0.17/-0.04 | 1.0/1.7 | 1.6/2.3 | 1.1/1.7 | 0.8/0.9 | 0.9/1.7 | 1.1/0.6 |
| R2 | 0.02/0.01 | 0.6/0.5 | 0.8/0.5 | 0.9/0.7 | 0.5/0.2 | 0.7/0.7 | 0.5/0.7 |
| MB | -0.03/-0.08 | -0.08/-0.14 | 0.01/-0.04 | 0.006/-0.05 | 0.03/-0.02 | 0.004/-0.05 | -0.09/-0.03 |
| CRMSD | 0.04/0.03 | 0.03/0.02 | 0.04/0.03 | 0.03/0.03 | 0.04/0.03 | 0.04/0.03 | 0.03/0.04 |
| **MAM** | | | | | | | |
| Slope | 0.4/0.36 | 0.6/0.4 | 0.8/0.6 | 0.6/0.8 | 0.5/0.3 | 0.9/0.7 | 0.6/0.6 |
| R2 | 0.2/0.08 | 0.3/0.04 | 0.5/0.15 | 0.9/0.5 | 0.3/0.14 | 0.4/0.2 | 0.2/0.4 |
| MB | -0.04/-0.08 | -0.08/-0.11 | 0.005/-0.03 | 0.003/-0.03 | 0.02/-0.03 | -0.02/-0.05 | -0.07/-0.03 |
| CRMSD | 0.03/0.03 | 0.03/0.03 | 0.03/0.03 | 0.03/0.03 | 0.04/0.03 | 0.03/0.03 | 0.03/0.03 |
| **JJA** | | | | | | | |
| Slope | 0.26/0.38 | 0.3/0.4 | 0.02/0.15 | 0.1/0.3 | 0.08/-0.04 | 0.05/0.06 | 0.3/0.1 |
| R2 | 0.02/0.01 | 0.04/0.005 | 0.001/0.002 | 0.8/0.17 | 0.003/0.012 | 0.01/0.003 | 0.01/0.01 |
| MB | -0.01/-0.03 | -0.04/-0.05 | 0.03/0.012 | 0.01/0.002 | 0.05/0.04 | 0.03/0.02 | -0.02/'-0.003 |
| CRMSD | 0.03/0.03 | 0.03/0.03 | 0.03/0.03 | 0.03/0.03 | 0.03/0.03 | 0.03/0.03 | 0.03/0.03 |
| **SON** | | | | | | | |
| Slope | 0.69/1.06 | 0.9/1.3 | 1.2/1.7 | 0.8/1.2 | 0.7/1.1 | 0.8/1.3 | 1.1/0.8 |
| R2 | 0.5/0.6 | 0.6/0.6 | 0.7/0.8 | 0.9/0.7 | 0.8/0.7 | 0.8/0.7 | 0.8/0.07 |
| MB | -0.02/-0.04 | -0.03/-0.05 | 0.04/-0.03 | 0.03/0.006 | 0.03/0.01 | 0.04/0.02 | -0.02/-0.003 |
| CRMSD | 0.04/0.04 | 0.04/0.04 | 0.04/0.04 | 0.04/0.04 | 0.04/0.04 | 0.04/0.04 | 0.04/0.04 |


















**Table 4:** Statistics of daily areal averages of ground-based SSM measurements in the OBS area versus point-like SURFEX-ISBA simulations at the same sites.


| | M-I | M-II | VAS | NIC | EZ | LC | OBS |
|---|---|---|---|---|---|---|---|
| **All period** | | | | | | | |
| Slope | 0.9 | 1.3 | 0.9 | 0.7 | 1.0 | 0.9 | 1.0 |
| R2 | 0.8 | 0.8 | 0.8 | 0.8 | 0.8 | 0.7 | 0.9 |
| MB | 0.004 | -0.012 | 0.011 | 0.006 | 0.02 | 0.006 | 0.005 |
| CRMSD | 0.02 | 0.02 | 0.02 | 0.02 | 0.01 | 0.02 | 0.02 |
| **DJF** | | | | | | | |
| Slope | 0.2 | 1.3 | 0.8 | 1.2 | 1.2 | 1.1 | 1.1 |
| R2 | 0.03 | 0.4 | 0.4 | 0.7 | 0.7 | 0.5 | 0.6 |
| MB | 0.01 | -0.03 | 0.02 | 0.03 | 0.02 | 0.03 | 0.01 |
| CRMSD | 0.04 | 0.05 | 0.03 | 0.04 | 0.03 | 0.03 | 0.04 |
| **MAM** | | | | | | | |
| Slope | 0.8 | 1.0 | 1.0 | 0.7 | 0.8 | 0.7 | 0.9 |
| R2 | 0.5 | 0.4 | 0.6 | 0.4 | 0.6 | 0.5 | 0.6 |
| MB | 0.002 | -0.02 | 0 | 0.01 | 0.01 | -0.02 | -0.004 |
| CRMSD | 0.04 | 0.02 | 0.03 | 0.04 | 0.03 | 0.04 | 0.04 |
| **JJA** | | | | | | | |
| Slope | 0.4 | 0.8 | 1.6 | 3 | 1.6 | 2 | 1.5 |
| R2 | 0.7 | 0.8 | 0.7 | 0.5 | 0.7 | 0.6 | 0.8 |
| MB | 0.004 | 0.01 | 0.01 | -0.02 | 0.02 | 0.005 | 0.005 |
| CRMSD | 0.04 | 0.02 | 0.03 | 0.04 | 0.03 | 0.04 | 0.04 |
| **SON** | | | | | | | |
| Slope | 0.9 | 1.1 | 0.9 | 0.8 | 1.0 | 1.1 | 1.0 |
| R2 | 0.8 | 0.8 | 0.8 | 0.9 | 0.9 | 0.8 | 0.9 |
| MB | 0.002 | 0 | 0.01 | 0 | 0.02 | 0.01 | 0.006 |
| CRMSD | 0.04 | 0.006 | 0.03 | 0.04 | 0.04 | 0.03 | 0.04 |



















**Figures**

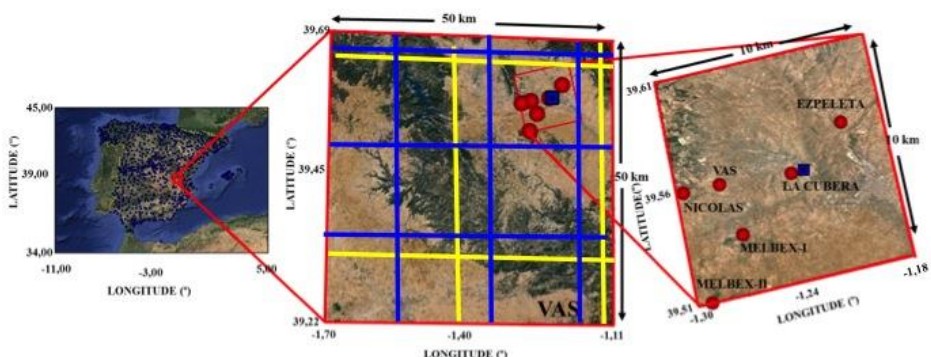



**Figure 1:** Area of investigation and orography. Location of rain gauges from AEMET
(Meteorological Service of Spain) is shown over the Iberian Peninsula (blue square dots).
The positions of the soil moisture network stations within the 10x10 km$^2$ (OBS area) in the
Valencia Anchor Station (VAS; 50x50 km$^2$) area are indicated by red circles.
















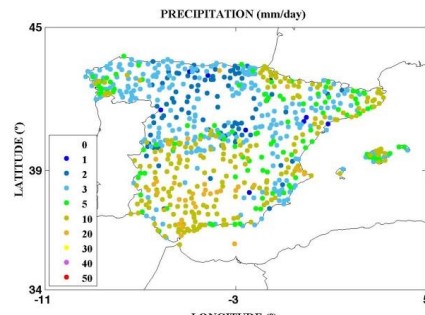


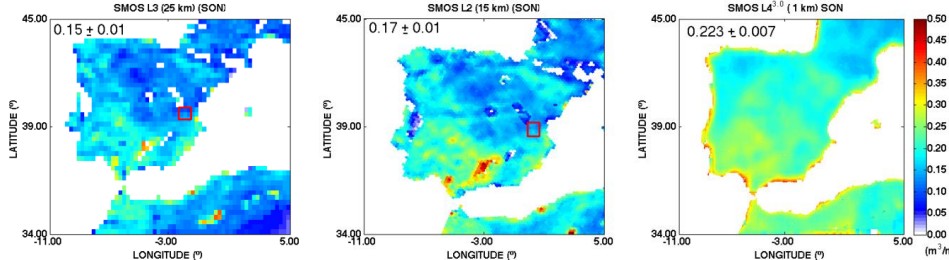



**Figure 2:** (a) Spatial distribution of precipitation over the Iberian Peninsula from the network of rain gauges of AEMET. The period of September to November (SON) 2012 is shown. (b) Spatial distribution of SMOS-derived soil moisture over the Iberian Peninsula (merged product: ascending and descending orbits, days with areal coverage higher than 50 % are considered).


















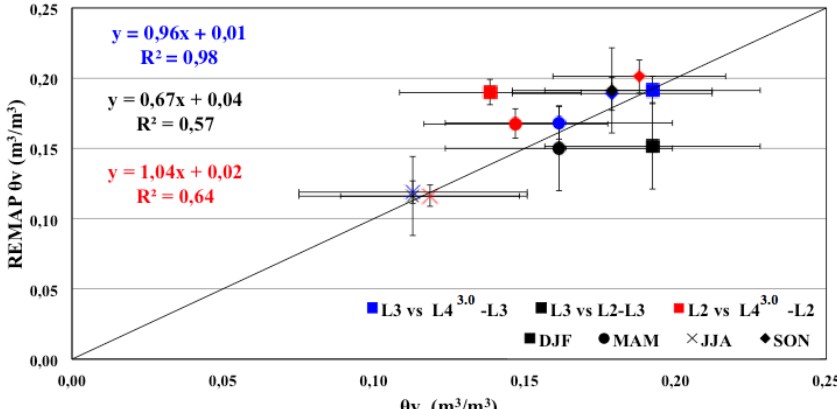



**Figure 3:** SMOS-derived SSM products comparison from different operational levels over the
Iberian Peninsula.



















(a)

(b)

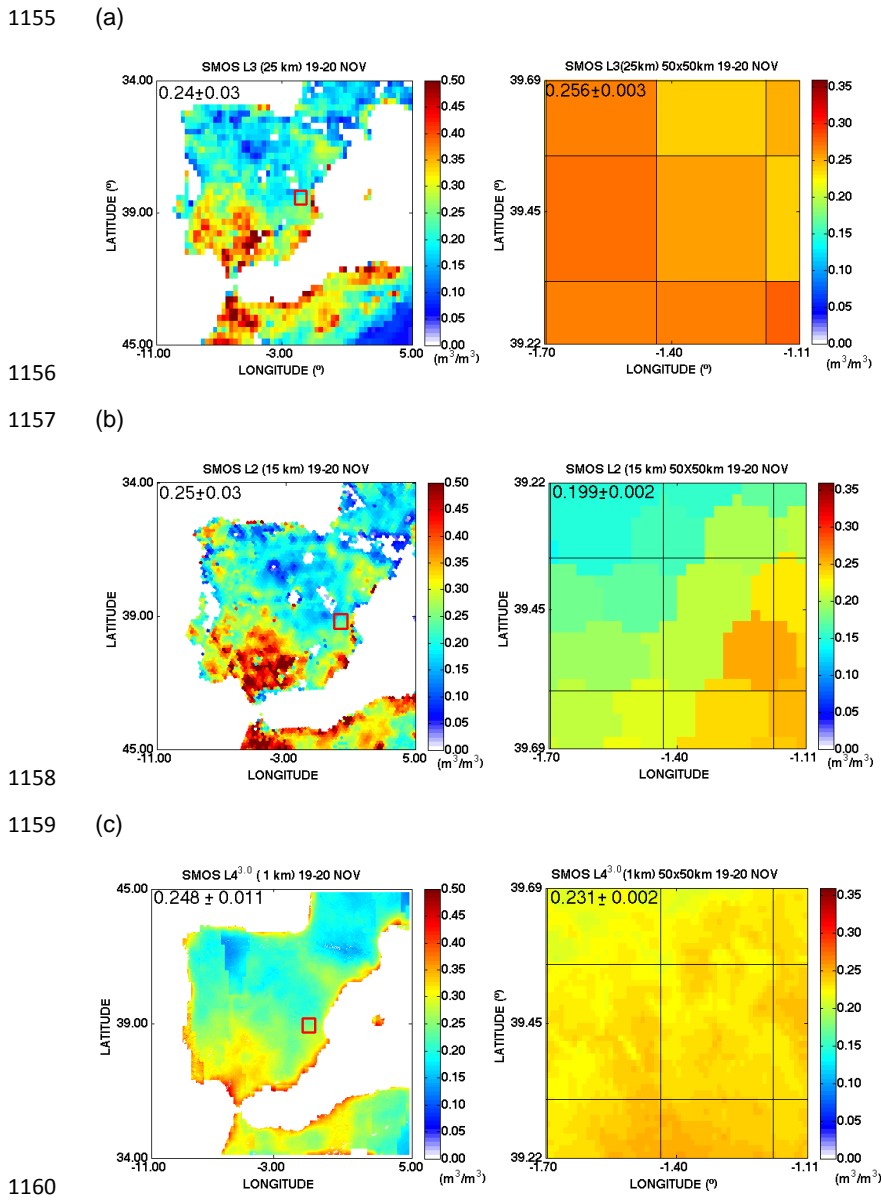


(c)


**Figure 4:** Spatial distribution of SMOS-derived soil moisture (merged product: ascending and descending orbits are considered) over the Iberian Peninsula (left) and the VAS (right) as a mean for the 19-20 November of 2012 (a) SMOS-L3 (~25 km), (b) SMOS-L2 (~15 km), (c) SMOS-L4[3.0] (~1 km). White empty pixels in (a) and (b) are indicative of a lack of data. Please be aware of the different colour scale used for the IP and VAS.






(a)

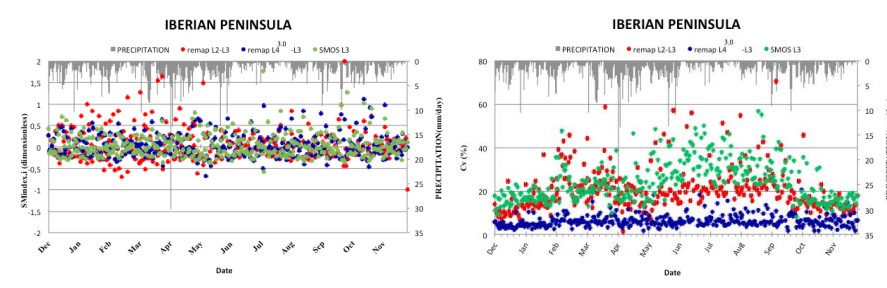



(b)

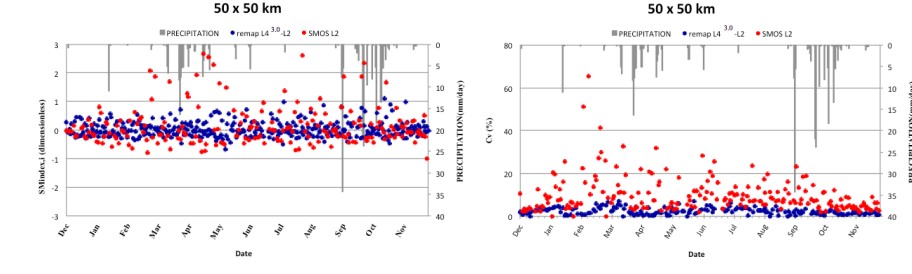


(c)

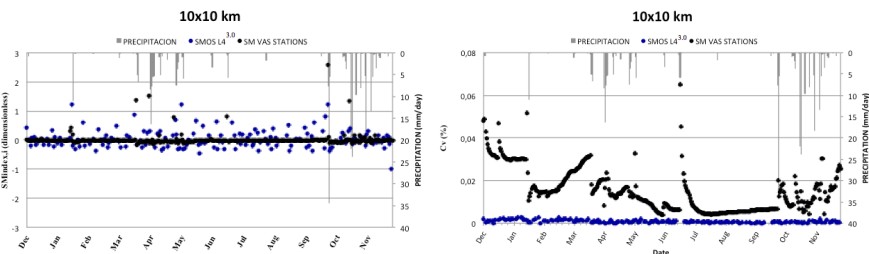


**Figure 5**: Averaged SMOS products and averaged ground-based observations of soil
moisture evolution over the Iberian Peninsula (IP; top), the VAS area (centre), and the OBS
area (bottom). Descending orbits are used. Precipitation from AEMET rain gauges on top.
Left) Soil moisture daily index ($\Theta_{v\,index,i}$; dimensionless) and right) Coefficient of variation (Cv,
1179     %).




(a)

(b)

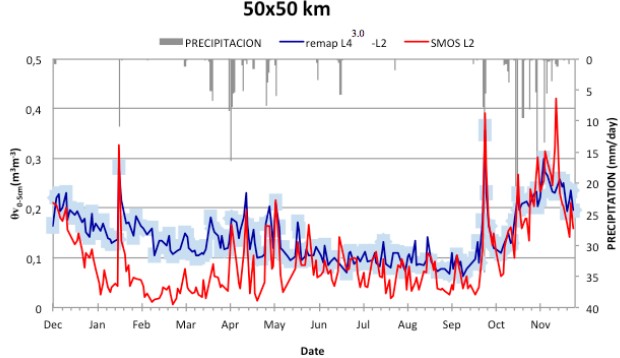


(c)

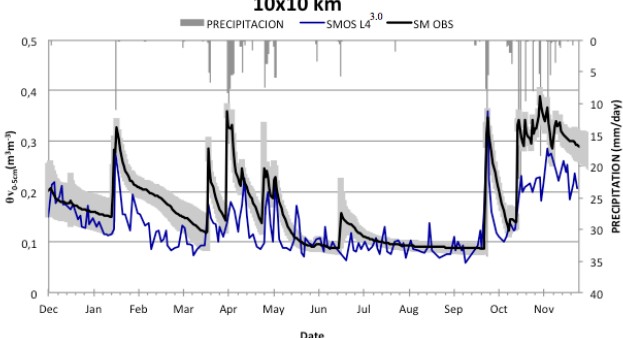


**Figure 6:** Temporal evolution of surface soil moisture time series averaged over the Iberian
Peninsula (top), the VAS area (50 x 50 km$^2$; centre) and the OBS area (10 x 10 km$^2$; bottom).
SMOS afternoon orbits are considered. Daily mean precipitation from the AEMET stations is
shown on top of each plot.  SMOS and remapped SMOS products are indicated in the plots.
Shaded areas show standard deviations, respectively.





(a)

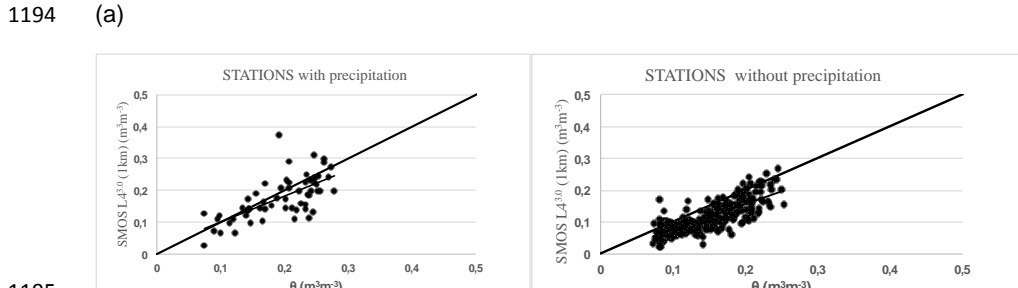



(b)

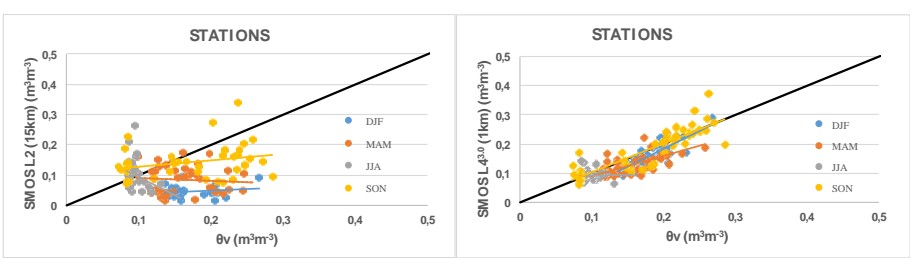



(c)

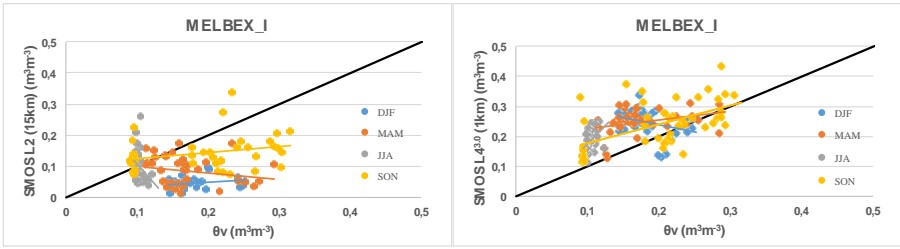



**Figure 7:** Results of the seasonal analysis for the hydrological year starting in December
2011. Scatter plots of (a) SMOS-L4$^{3.0}$ SSM (ascending and descending orbits) versus
averaged 10x10 km$^2$ in situ soil moisture measurements (left) for days with precipitation, and
(right) and without precipitation (< 1 mm /d). (b) SMOS-L2 and SMOS-L4$^{3.0}$ SSM (descending
orbits) versus averaged 10x10 km$^2$ in situ soil moisture measurements. (c) SMOS-L2 and
SMOS-L4$^{3.0}$ SSM (descending orbits) versus point-like ground measurements from
MELBEX_I station, using the closest grid point. Segments are linear fit of seasonal data (3
months data). Statistics for individual comparisons at all stations are summarized in Table 3.





(a)

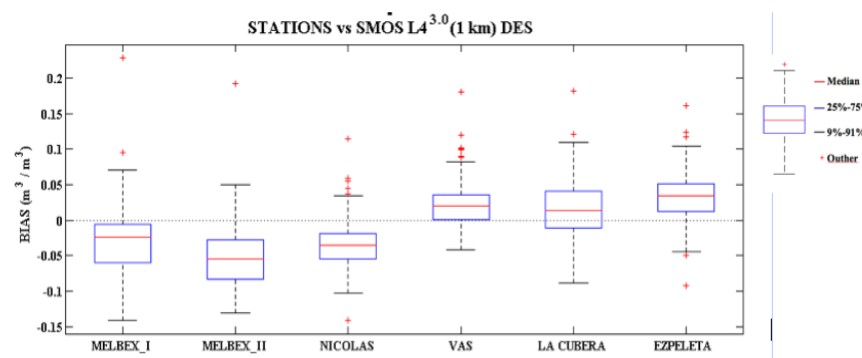



(b)

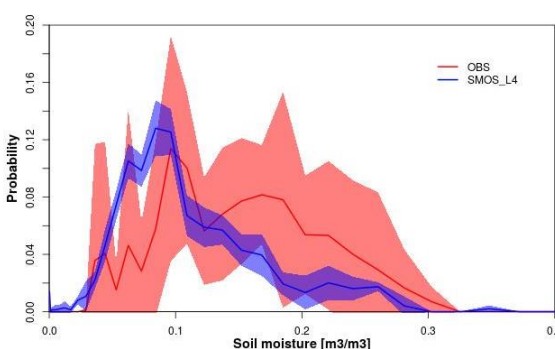


**Figure 8:** (a) Box plot of the comparison between point-like ground measurements at all
stations over the VAS area and closest SMOS-L4$^{3.0}$ SSM data. (b) Probability distribution
funtion (PDF) of SSM from in situ observations and SMOS- L4$^{3.0}$ SSM measurements. The
standard deviations are indicated with shaded areas. Full lines represent the mean over all
ground stations and over the 10 x 10 km$^2$ of the OBS area in VAS where the in SSM network
is located.








(a)

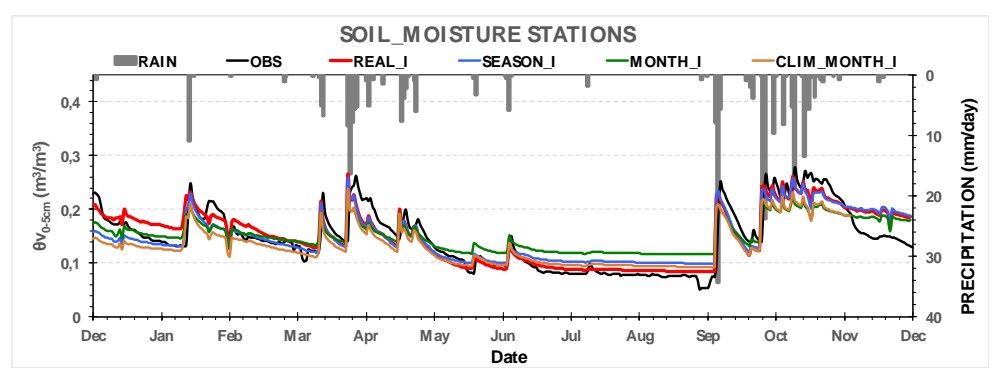


(b)

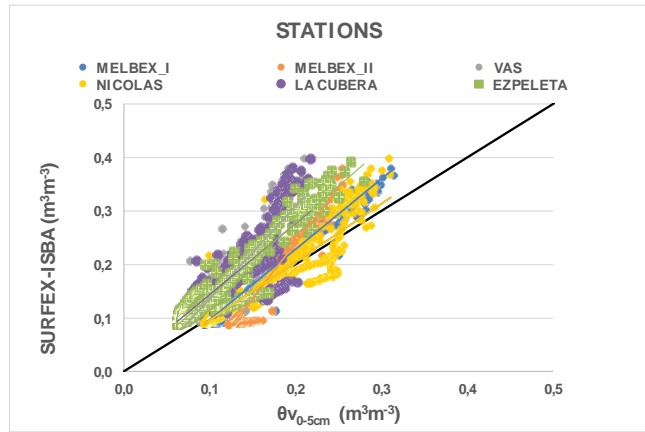



**Figure 9**: (a) Temporal evolution of SSM in situ measurements and simulated SURFEX-
ISBA as a mean over all stations. All perturbation simulations are indicated. Precipitation
from AEMET stations is included at the top. (b) Scatter plot of temporal mean (over the whole
simulation period) SSM ground measurements versus SURFEX-ISBA simulations (realistic
initial scenario; REAL-I) at all stations. Statistics for all stations using the REAL-I initial
scenario are presented in Table 4.









(a)

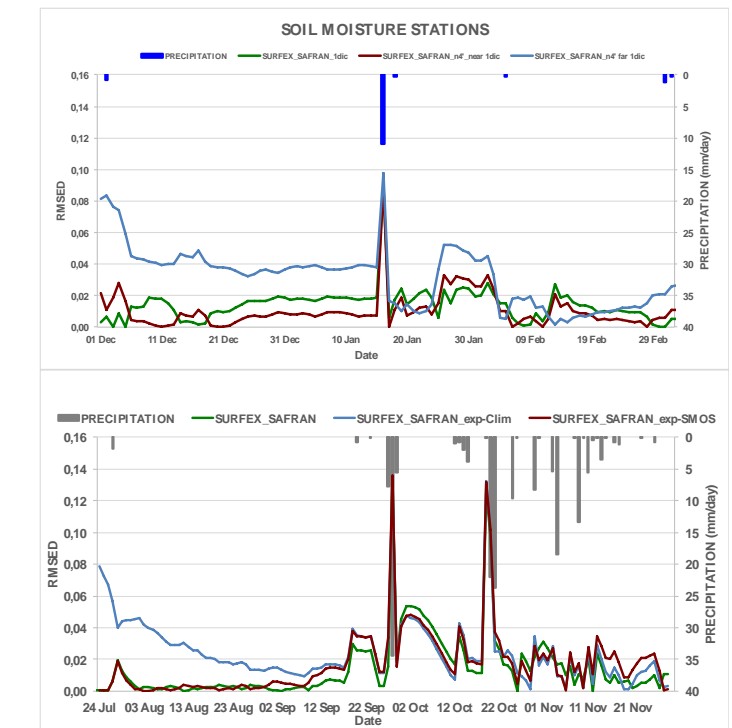




(b)

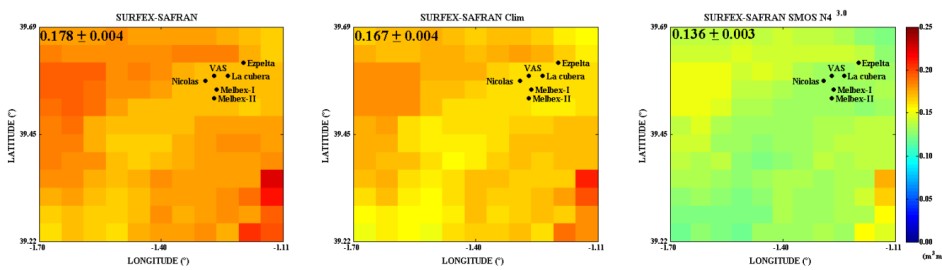








(c)

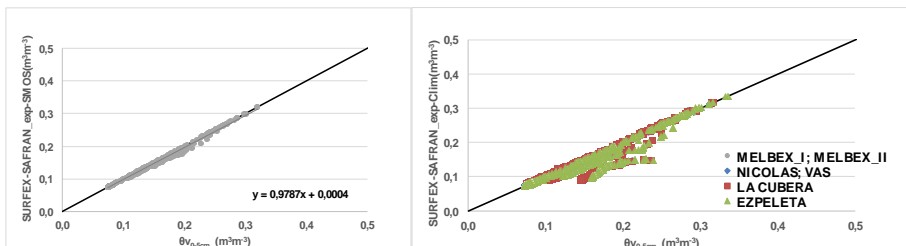




**Figure 10:** (a) RMSD for the daily mean SSM from the three SURFEX-ISBA simulations with perturbed initial SSM scenarious (details in section 4.3.2). (b) Spatial distribution of mean SSM for the winter simulation (a, left) for the 3 simulations. (c) Scatter plot depicting the compariosn between in situ SSM observations and SURFEX-SAFRAN-SMOSL4[3.0] simulations, as a mean over all stations (left) and for each of the stations (right).



