# Peer review of "An improved perspective in the spatial representation of soil moisture: potential added value of SMOS disaggregated 1 km resolution "all weather" product"

_Hydrology and Earth System Sciences, 2018_

## Referee Comment (RC1) · Anonymous Referee #1 · 19 Feb 2018

Review of "An improved perspective in the representation of soil moisture : potential and added value of SMOS disaggregated 1 km resolution product" by Khodayar et al.

This study attempts to assess the potential of a new SMOS surface soil moisture (SSM) product to initialize ISBA land surface model, previously implemented at fine scale over the VAS (Valencia Anchor Station) area. It notably represents the first application of the newly released SMOS level 4 "all weather" disaggregated 1 km soil moisture, based on the ERA-Interim Land surface temperature (LST).

Although the study is interesting, the manuscript is a little bit confusing, and the structure should be better organized. Moreover, it is unclear why the authors chose to

aggregate the 1 km resolution product if the final objective is to improve the spatial representation of soil moisture. An in-depth analysis of the SMOS level 4 data and its added value compared to Level 2 or Level 3 data is required since no reference is given elsewhere.

Please find below a list of major comments, followed by specific points.

Major comments:

1) Few information about the SMOS L4 version 3.0 (section 2.2) are given and the reference Piles et al. 2015 (Quality report) could not be found in the reference list. When looking at SMOS L4 data maps (Figures 2 and 4), one question arises strikingly: what is the actual spatial resolution of the downscaled SSM? The spatial resolution of SMOS L4 seems to be much larger than that of L2 and L3. Has the meteorological forcing used to derive ERA-Interim LST anything to do with the apparent resolution of L4 product? What is the spatial resolution of ERA-Interim LST?

2) Another concern with the use of ERA-Interim LST data for downscaling SMOS data. As the LST is derived numerically from the ERA-Interim soil moisture data via the energy budget model of TESSEL, would it be equivalent to use the ERA-Interim soil moisture data directly?

3) Evaluation of the SSM product: Line 366: "the higher resolution SMOS L4 showing lower standard deviation". Line 415: "The CVs of the spatially averaged SMOS L4 is lower than those of SMOS L3 and L2 and in situ observations indicating that this data are less scattered." In my opinion, a lower variability for the downscaled SSM product is unexpected. It should be the opposite: higher variability for the downscaled SSM. Line 393: "L4 product shows SSM mean and variability in the same range of the SMOS L2 and L3 products, but with a finer-improved resolution representation of the spatial distribution". L398: "the potential added value of the 1 km product is manifest". The SMOS L4 has a spatial variability much lower than that of both L2 and L3 products. How to demonstrate that the slight 1 km variability is real information

and not an artifact (oversampling)? Line 633: "consistent with the finer resolution of this product which better captures local information on the 1 km x 1 km pixel, whereas coarser products smooth out this vital information". To me, there is no information in this paper supporting the hypothesis that the downscaled product improves the spatial representation of SMOS L2 and L3 products. To really evaluate the SMOS L4 product, one should compare (in Table 3 for instance) the SMOS L4 versus in situ and SMOS L2 (or L3) versus in situ for each station separately, that is at a scale finer than the L2/L3 spatial resolution. Are statistics better for L4 than for L2 or L3? Bottom subtable of Table 3 is unclear. In addition errors are identified in the right column (OBS), which does not always correspond to the mean for all stations (?).

4) In the present form, the paper is a bit lengthy. The description of approaches is sometimes repetitive. The structure of the manuscript could be improved. For instance: lines 334-335 (and lines 507 to 512) three to four initialization experiments are presented, but the initialization using SMOS data is not mentioned, although claimed as the main objective of the paper. Conclusions are confusing as well. The authors should better highlight their findings by selecting few key results.

5) As the study focuses on SMOS derived SSM at high spatial temporal resolution including all weather conditions, I suggest two recent references to complement the state-of-the-art presented in the introduction: Malbéteau, Y., Merlin, O., Balsamo, G., Er-Raki, S., Khabba, S., Walker, J. P., Jarlan, L. (2018). Toward a Surface Soil Moisture Product at High Spatiotemporal Resolution: Temporally Interpolated, Spatially Disaggregated SMOS Data. Journal of Hydrometeorology, 19(1), 183-200. Djamai, N., Magagi, R., Goïta, K., Merlin, O., Kerr, Y., Roy, A. (2016). A combination of DISPATCH downscaling algorithm with CLASS land surface scheme for soil moisture estimation at fine scale during cloudy days. Remote Sensing of Environment, 184, 1-14.

6) Line 529: "soil moisture initialization in spatialized SURFEX simulations requires a single representative value for the whole simulation area. In this case, we use as input the SMOS L4 1 km disaggregated soil moisture mean over the whole simulation

area for the initialization day". Why not initializing the model at 1 km resolution if 1 km resolution data are available? What is the point of disaggregating SMOS L2/L3 data then?

7) On the usefulness of surface soil moisture data to initialize ISBA. Line 229: "Particularly relevant for this study is the specific definition of the soil hydraulic parameters which they made for the VAS area, since most of the hydrological parameters are site dependent". Does the approach require in situ measurements for the calibration? Since the objective is to initialize ISBA using SMOS L4 data, I am wondering whether the site specific calibration could be done using SMOS L4 data solely (without relying on in situ measurements for ISBA simulations). Line 488: "Initialization of land surface models is a crucial issue and its impact on the accuracy of model estimation is widely recognized to be significant". What about the initialization of the root-zone soil moisture, which has supposedly more weight in the initialization than the SSM?

Specific points:

- It is unclear at which spatial resolution ISBA model is run over the VAS?

- Confusion is often made between observation and sampling grid resolution. Ex. Line 10: 25 km and 15 km are the resolutions of sampling grids, the actual spatial resolution for both products being about 40 km.

- Figure 2 (and Figure 4): Image at the middle is not correctly georeferenced compared to the left (top) and right (bottom) images.

- Units in m3/m3 are sometimes missing the text and the figures.

- Line 306: "SMOS L4 soil moisture grid cells are averaged over the 10x10 km2 area and compared to the mean from the soil moisture network stations to address the issue related to spatial averaging". Please clarify the issue to be addressed?

- Notations: SURFEX-SAFRAN (SURFEX forced by SAFRAN), SURFEX-ECMWF (SURFEX forced by ERA-Interim) and SURFEX-ISBA are used. The terminology

SURFEX-ISBA is confusing as it corresponds to SURFEX (ISBA) forced by station-based meteorological measurements. For clarity, I suggest to replace SURFEX-ISBA by (for instance) SURFEX-VAS

- Some references are missing in the reference list: I have noted Louvet et al. 2015; Piles et al. 2015; and maybe others.

---

## Referee Comment (RC2) · Anonymous Referee #2 · 3 Mar 2018

**OVERVIEW**

The manuscript describes the potential added value of the "all weather" disaggregated 1-km SMOS L4 (v3.0) soil moisture product obtained by integrating SMOS, MODIS (optical) and Land Surface Temperature data by ERA-Interim. Indeed, by using ERA-Interim modelled data the problem of missing observations during cloudy conditions is overcome. The comparison with SMOS L2 and L3 (CATDS) soil moisture product over the Iberian Peninsula, and specifically over the VAS area, is carried out. Moreover,

the initialization of the SURFEX-ISBA model with SMOS-derived soil moisture data to assess the impact of the model initialization on the simulations is investigated.

**GENERAL COMMENTS**

The manuscript investigates a relevant topic. The recent availability of 1-km soil moisture products from the disaggregation of coarse resolution retrievals, and from high-resolution microwave sensors (e.g., Sentinel-1), still need to be thoroughly assessed and, particularly, tested the potential added value in hydrological or climatic applications. By reading the title, I was really interested to the paper and I thought its content was different with respect to the current text. I expected a more general view in which the added value of the high resolution product in real-world application(s) was determined. Therefore, I firstly suggest changing the title that is misleading.

I have to admit that I couldn't resist to read the comments of reviewer 1 who made a very good review and I agree with most of her/his comments (that should be addressed). Additionally, I have three major comments:

1) The paper is too long, not well organized (e.g., several repetitions), and not focused to a clear message. The new SMOS L4 (v3.0) "all weather" product is introduced. However, a little description of the product is carried out, with a reference to a "Quality Report" not present in the reference list. As highlighted by reviewer 1, many details are missing (e.g., spatial resolution of ERA-Interim LST, its merging with MODIS-derived LST, . . .). These points need to be clarified. The title should be changed to underline the presentation of the new product. The whole paper should be focused on this new product.

2) More important than point 1, the paper should be focused clearly on the more relevant aspects the authors want to convey to the readers. The disaggregated product has a spatial resolution of 1-km, the assessment should be carried out with observations and/or modelling at 1-km resolution. It is not done in the paper. As in most "soil moisture downscaling papers" the assessment of the disaggregated product is carried out in the *TEMPORAL DOMAIN*, usually concluding that as the disaggregated product shows similar performance than the coarse resolution product. Being at higher resolution, it is a better product. Unfortunately, for me it is wrong and misleading.

I expected that the new disaggregated product was compared with high resolution modelled data (constrained by in situ observations) in the *SPATIAL DOMAIN*. This comparison is needed to understand if the disaggregated product is able to reproduce the high resolution soil moisture variability (at 1-km scale). Of course, the model should be forced with high resolution meteorological forcing (e.g., radar rainfall), and it is hard to be done.

In my opinion, the comparison with SMOS L2 and L3 products should be strongly reduced and the authors should focus on the SPATIAL assessment of the SMOS L4 "all weather" product (likely compared with SMOS L4 v2 product not including ERA-Interim LST). If the new product is able to reproduce the spatial variability of high resolution modelled data, then the authors can say that "the SMOS L4 v3 product captures the 1-km soil moisture spatial variability". Otherwise, all the sentences similar to this one should be removed by the paper.

3) The analysis for the initialization of modelled data is, at least for me, not clear and likely not appropriate. To assess the added value of the soil moisture product, the authors should introduce the product into the modelling (e.g., through data assimilation) and assess the model performance without and with the use of the product. Specifically, the authors should assimilate different SMOS products into the modelling and then assess the best product based on the simulation results after the assimilation. The authors only showed that if different initial soil moisture conditions are considered,

different results are obtained. However, this is highly expected and largely shown in the scientific literature. An assimilation analysis I guess goes beyond the scope of the paper. Therefore, I am suggesting removing, or strongly reducing, this part.

Some specific comments and corrections should be also addressed. For instance, the introduction introduces ONLY SMOS among the satellite soil moisture products currently available. We have SMAP, ASCAT, AMSR2, ESA CCI and Sentinel-1 as operational products freely available. They should be at least mentioned.

Some of the comments are already reported by reviewer 1 and I believe the paper in the current form should be significantly modified. Hence, I have not included the specific corrections at this stage.

**RECCOMMENDATION**

On this basis, I found the topic of the paper relevant, but the paper still needs substantial improvements. Therefore, I suggest a major revision before the possible publication in Hydrology and Earth System Sciences.
* * *

---

## Author Comment (AC1) · 4 May 2018

**Answers to Reviewer 2**

We thank reviewer 2 for all his/her suggestions. All of them will be considered in detail for the correction phase of the manuscript. In the following a general description of the main changes to be applied and detail answers to the comments is presented.

Kind regards,
Samiro Khodayar on behalf of all co-authors.
* * *
In the following a description of the main changes suggested is summarized,
- Proposed title change:
    An improved perspective in the representation of soil moisture: potential added value of SMOS disaggregated 1 km resolution "all weather" product

Better definition of the objective, novelty and relevance of this study improving the structure, content and length of the publication accordingly:

1. To examine the benefits of the SMOS L4 version 3.0 or "all weather" high resolution soil moisture disaggregated product (~ 1 km, $SMOS\_L4^{3.0}$).
    o *The added value compared to SMOS-L3 (~ 25 km) and L2 (~15 km) is investigated.*
    o *High-temporal (every 10 min over several years) and spatial (7 stations in an area of about 10 x 10 km) soil moisture observations from the Valencia Anchor Station (VAS; SMOS Calibration/Validation (Cal/Val) site in Europe) are used for comparison and assessment of the spatio-temporal performance of the satellite derived soil moisture products.*
    o *The SURFEX-ISBA model is used to simulate point-scale surface SM (SSM) and, in combination with high-quality atmospheric information data, namely ECMWF and the SAFRAN meteorological analysis system, to obtain a representative SSM mapping over the VAS.*
    2. First study, to the authors knowledge, apart from the quality report, that makes use of the newly SMOS L4 3.0 "all weather" soil moisture product.

- *Added value compared to Level 2 and 3 SMOS products*
- *Validation of the $SMOS\_L4^{3.0}$ product in a different climatic region than REMEDHUS (Quality Report, Piles et al 2015)*
- *Temporal and spatial assessment of the performance of the $SMOS\_L4^{3.0}$ product including a seasonal analysis*
- *First examples of possible applications of this product for initialization of off-line Soil-Vegetation-Atmosphere Transfer models (in this case SURFEX-ISBA) in stand-alone or regional approaches.*

3. The comparison carried out helps drawing guidelines on best practices for the sensible use of these products. Currently, there is not a consensus about what is the "best" SMOS product. Different users utilize different products depending on their application rather than based on performance arguments. This study and the conclusions obtained on the comparison are important to provide information on the advantages and drawbacks of these datasets.

Furthermore, regional SM maps with high accuracy are needed for flood forecasting, crop monitoring and crop development strategies, among others. Correct initial conditions for model simulations of these SM maps are fundamental to obtain a good accuracy. SMOS-L4$^{3.0}$ could fill the actual information gap and fulfil this requirement.

- New references have been included following the reviewers suggestions:

  o Piles, M., Pou, X., Camps, A., Vall-llosera, M. (2015): Quality report: Validation of SMOS-BEC L4 high resolution soil moisture products, version 3.0 or "all-weather". Technical report. Available at: http://bec.icm.csic.es/doc/BEC-SMOS-L4SMv3-QR.pdf

  o SMOS-BEC Team (2016): SMOS-BEC Ocean and Land Products Description. Technical report. Available at: http://bec.icm.csic.es/doc/BEC-SMOS-0001-PD.pdf

  o Malbéteau, Y., Merlin, O., Balsamo, G., Er-Raki, S., Khabba, S.,Walker, J. P., Jarlan, L. (2018). Toward a Surface Soil Moisture Product at High Spatiotemporal Resolution: Temporally Interpolated, Spatially Disaggregated SMOS Data. Journal of Hydrometeorology, 19(1), 183-200.

  o Djamai, N., Magagi, R., Goïta, K., Merlin, O., Kerr, Y., Roy, A. (2016). A combination of DISPATCH downscaling algorithm with CLASS land surface scheme for soil moisture estimation at fine scale during cloudy days. Remote Sensing of Environment, 184, 1-14.

  o Louvet, S., Thierry Pellarin, Ahmad al Bitar, Bernard Cappelaere, Sylvie Galle, Manuela Grippa, Claire Gruhier, Yann Kerr, Thierry Lebel, Arnaud Mialon, Eric Mougin, Guillaume Quantin, Philippe Richaume, Patricia de Rosnay (2015). SMOS soil moisture product evaluation over West-Africa from local to regional scale. Remote Sensing of Environment, Volume 156, Pages 383-394, ISSN 0034-4257, DOI: 10.1016/j.rse.2014.10.005.

**GENERAL COMMENTS**

1) **The manuscript investigates a relevant topic. The recent availability of 1-km soil moisture products from the disaggregation of coarse resolution retrievals, and from high resolution microwave sensors (e.g., Sentinel-1), still need to be thoroughly assessed and, particularly, tested the potential added value in hydrological or climatic applications. By reading the title, I was really interested to the paper and I thought its content was different with respect to the current text. I expected a more general view in which the added value of the high resolution product in real-world application(s) was determined. Therefore, I firstly suggest changing the title that is misleading.**

The main goal of this study is to investigate the added value of the 1 km "all weather" product with respect to coarser resolutions, the SMOS-L3 (~ 25 km) and L2 (~15 km) products, undergoing an evaluation against in situ observations. Additionally, in a first simple approach examples of possible applications of this product for initialization of off-line Soil-Vegetation-Atmosphere Transfer models (in this case SURFEX-ISBA) in stand-alone or regional approaches are presented.

As described for the reviewer 1, in a new study of the first author, which is about to be submitted to HESS, the suggestion of the reviewers is explored, in which we assess the benefit of using the SMOS-L4 product for the initialization of high-resolution convective-permitting simulations to improve the predictability of extreme weather phenomena such as heavy precipitation.

We suggest to slightly modify the title: An improved perspective in the representation of soil moisture: potential added value of SMOS disaggregated 1 km resolution "all weather" product, to better reflect which product we refer to, as suggested by the reviewer.

**Major comments:**
**The paper is too long, not well organized (e.g., several repetitions), and not focused to a clear message.**
We will follow the reviewer's suggestion and try to remove all repetitions and better describe the main goals/focus of this study.

**The new SMOS L4 (v3.0) "all weather" product is introduced. However, a little description of the product is carried out, with a reference to a "Quality Report" not present in the reference list. As highlighted by reviewer 1, many details are missing (e.g., spatial resolution of ERA-Interim LST, its merging with MODIS-derived LST, . . .). These points need to be clarified. The title should be changed to underline the presentation of the new product. The whole paper should be focused on this new product.**
The references to the quality report as well as other publications of relevance to the topic have been included in the reference list. Additional information regarding details of the SMOS L4 3.0 product which could be helpful for the reader will be included in the text. The title has been slightly modified to better identify the product we are discussing. We do not intend to introduce the new SMOS L4 (v3.0) "all weather" product, which is not ours (it was developed at BEC as described in the manuscript), but just to show the added value of the product with respect to other SMOS-derived SM products and give a simple example of the potential benefit of the new product.

**2) More important than point 1, the paper should be focused clearly on the more relevant aspects the authors want to convey to the readers. The disaggregated product as a spatial resolution of 1-km, the assessment should be carried out with observations and/or modelling at 1-km resolution. It is not done in the paper. As in most "soil moisture downscaling papers" the assessment of the disaggregated product is carried out in the TEMPORAL DOMAIN, usually concluding that as the disaggregated product shows similar performance than the coarse resolution product. Being at higher resolution, it is a better product. Unfortunately, for me it is wrong and misleading. I expected that the new disaggregated product was compared with high resolution modelled data (constrained by in situ observations) in the SPATIAL DOMAIN. This comparison is needed to understand if the disaggregated product is able to reproduce the high resolution soil moisture variability (at 1-km scale). Of course, the model should be forced with high resolution meteorological forcing (e.g., radar rainfall), and it is hard to be done.**

The spatio-temporal correlations are analysed through comparison with point-scale observations over the VAS region. A network of six stations is located in an area of about 10x 10 km$^2$. Section 4.2, lines 438 to 477, is devoted to the comparison of SMOS L4 and –L2 products to the in situ measurements from the VAS network. Statistics for individual comparisons at all stations are summarized in Table 3. Figures 7, 8 and even 9 are devoted to these comparisons, although it is not possible to always show all stations due to space issues. In the description, details are given about the better accuracy of –L4 product. An assessment of the quality of the SMOS L4 product using high resolution modelled data, even when constrained by in situ observations, is not a correct approach since modelled data present relevant biases. In general, the observations, as used in this study are considered "the truth"; hence, they are used for validation of satellite products. Indeed, when for example soil moisture products are used for initialization and/or assimilation in our models the correct approach is to apply CDF (Cumulative Distribution Function) matching methodology to similarly rescale both products.

**In my opinion, the comparison with SMOS L2 and L3 products should be strongly reduced and the authors should focus on the SPATIAL assessment of the SMOS L4 "all weather" product (likely compared with SMOS L4 v2 product not including ERA-Interim LST). If the new product is able to reproduce the spatial variability of high resolution modelled data, then the authors can say that "the SMOS L4 v3 product captures the 1-km soil moisture spatial variability". Otherwise, all the sentences similar to this one should be removed by the paper.**

We agree with reviewer 1 that an analysis of the SMOS level 4 data and its added value compared to Level 2 or Level 3 data is interesting since no reference is given elsewhere. The comparison carried out helps drawing guidelines on best practices for the sensible use of these products. Different users utilize different products depending on their application rather than based on performance arguments. This study and the conclusions obtained on the comparison are important to provide information on the advantages and drawbacks of these datasets. Nevertheless, following the reviewer's suggestion we will reduce this part and only focus on the most relevant information, always reinforcing the role of the SMOS L4 3.0 product.
Concerning the comparison with the SMOS L4 2.0 product, the comparison was made during our analysis but results were not included in this manuscript, but following the reviewer's suggestion we will describe in the text the most relevant conclusions obtained from this comparison.

**3) The analysis for the initialization of modelled data is, at least for me, not clear and likely not appropriate. To assess the added value of the soil moisture product, the authors should**

**introduce the product into the modelling (e.g., through data assimilation) and assess the model performance without and with the use of the product. Specifically, the authors should assimilate different SMOS products into the modelling and then assess the best product based on the simulation results after the assimilation. The authors only showed that if different initial soil moisture conditions are considered, different results are obtained. However, this is highly expected and largely shown in the scientific literature. An assimilation analysis I guess goes beyond the scope of the paper. Therefore, I am suggesting removing, or strongly reducing, this part.**

As the reviewer correctly points out a data assimilation exercise was not the goal of this study and it was out of the scope of this paper. The problematic associated with the initialization of soil moisture in model simulations across scales is also a well-known and still a hot topic that deserves further consideration. As the reviewer pointed out "if different initial soil moisture conditions are considered, different results are obtained", in our first initialization exercise we wanted to stress this point out and assess the potential change that could be expected when different "normally" used initialization values are used. In the second part of the analysis, an initialization exercise using SMOS L4 3.0 information is presented. Following the reviewer's suggestion we will reduce this part and better clarify our purpose and results.

**Some specific comments and corrections should be also addressed. For instance, the introduction introduces ONLY SMOS among the satellite soil moisture products currently available. We have SMAP, ASCAT, AMSR2, ESA CCI and Sentinel-1 as operational products freely available. They should be at least mentioned.**

We agree with the reviewer and we will include in the introduction additional information regarding other operational products freely available.

---

## Author Comment (AC2) · 8 May 2018

**Answers to Reviewer 1**

We thank reviewer 1 for all his/her suggestions and comments. We considered all of them and will modify the manuscript accordingly. In the following, one may find a general description of the main changes to be applied and detail answers to his/her comments.

Kind regards,
Samiro Khodayar on behalf of all co-authors.
* * *
In the following a description of the main changes suggested is summarized,

- Proposed title change:
  An improved perspective in the representation of soil moisture: potential added value of SMOS disaggregated 1 km resolution "all weather" product

- Better definition of the objective, novelty and relevance of this study improving the structure, content and length of the publication accordingly:
  1. To examine the benefits of the SMOS L4 version 3.0 or "all weather" high resolution soil moisture disaggregated product (~ 1 km, SMOS_L4$^{3.0}$).
     o *The added value compared to SMOS-L3 (~ 25 km) and L2 (~15 km) is investigated.*
     o *High-temporal (every 10 min over several years) and spatial (7 stations in an area of about 10 x 10 km$^2$) soil moisture observations from the Valencia Anchor Station (VAS; SMOS Calibration/Validation (Cal/Val) site in Europe) are used for comparison and assessment of the spatio-temporal performance of the satellite derived soil moisture products.*
     o *The SURFEX-ISBA model is used to simulate point-scale surface SM (SSM) and, in combination with high-quality atmospheric information data, namely ECMWF and the SAFRAN meteorological analysis system, to obtain a representative SSM mapping over the VAS.*

     2. First study, to the authors knowledge, apart from the quality report, that makes use of the newly SMOS L4 3.0 "all weather" soil moisture product.

     - *Added value compared to Level 2 and 3 SMOS products*
     - *Validation of the SMOS_L4$^{3.0}$ product in a different climatic region than REMEDHUS (Quality Report, Piles et al 2015)*
     - *Temporal and spatial assessment of the performance of the SMOS_L4$^{3.0}$ product including a seasonal analysis*
     - *First examples of possible applications of this product for initialization of off-line Soil-Vegetation-Atmosphere Transfer models (in this case SURFEX-ISBA) in stand-alone or regional approaches.*

3. The comparison carried out helps drawing guidelines on best practices for the sensible use of these products. Currently, there is not a consensus about what is the "best" SMOS product. Different users utilize different products depending on their application rather than based on performance arguments. This study and the conclusions obtained on the comparison are important to provide information on the advantages and drawbacks of these datasets.

Furthermore, regional SM maps with high accuracy are needed for flood forecasting, crop monitoring and crop development strategies, among others. Correct initial conditions for model simulations of these SM maps are fundamental to obtain a good accuracy. SMOS-L4$^{3.0}$ could fill the actual information gap and fulfil this requirement.

- New references have been included following the reviewers suggestions:

  o Piles, M., Pou, X., Camps, A., Vall-llosera, M. (2015): Quality report: Validation of SMOS-BEC L4 high resolution soil moisture products, version 3.0 or "all-weather". Technical report. Available at: http://bec.icm.csic.es/doc/BEC-SMOS-L4SMv3-QR.pdf

  o SMOS-BEC Team (2016): SMOS-BEC Ocean and Land Products Description. Technical report. Available at: http://bec.icm.csic.es/doc/BEC-SMOS-0001-PD.pdf

  o Malbéteau, Y., Merlin, O., Balsamo, G., Er-Raki, S., Khabba, S.,Walker, J. P., Jarlan, L. (2018). Toward a Surface Soil Moisture Product at High Spatiotemporal Resolution: Temporally Interpolated, Spatially Disaggregated SMOS Data. Journal of Hydrometeorology, 19(1), 183-200.

  o Djamai, N., Magagi, R., Goïta, K., Merlin, O., Kerr, Y., Roy, A. (2016). A combination of DISPATCH downscaling algorithm with CLASS land surface scheme for soil moisture estimation at fine scale during cloudy days. Remote Sensing of Environment, 184, 1-14.

  o Louvet, S., Thierry Pellarin, Ahmad al Bitar, Bernard Cappelaere, Sylvie Galle, Manuela Grippa, Claire Gruhier, Yann Kerr, Thierry Lebel, Arnaud Mialon, Eric Mougin, Guillaume Quantin, Philippe Richaume, Patricia de Rosnay (2015). SMOS soil moisture product evaluation over West-Africa from local to regional scale. Remote Sensing of Environment, Volume 156, Pages 383-394, ISSN 0034-4257, DOI: 10.1016/j.rse.2014.10.005.

**Major comments:**

**1) Few information about the SMOS L4 version 3.0 (section 2.2) are given and the reference Piles et al. 2015 (Quality report) could not be found in the reference list.**

The quality report reference Piles et al. (2015) has been included in the reference list, thanks for noticing. Additionally, we included another reference to a document from the Barcelona Expert Center (BEC) with detailed information about all the products generated by BEC (SMOS-BEC Team (2016)). Unfortunately, after careful literature review no more references or information related to this product could be found. Nevertheless, some additional useful information has been included in the text, which can be found in the following question-answer. For further details regarding this product the SMOS BEC team should be contacted directly using the email address that is made available in the quality report. This information has been included in the manuscript.

**When looking at SMOS L4 data maps (Figures 2 and 4), one question arises strikingly: what is the actual spatial resolution of the downscaled SSM? The spatial resolution of SMOS L4 seems to be much larger than that of L2 and L3. Has the meteorological forcing used to derive ERA Interim LST anything to do with the apparent resolution of L4 product? What is the spatial resolution of ERA-Interim LST?**

The Level 4 SM, SMOS-L4 2.0 data (SMOS-L4$^{2.0}$), with 1 km spatial resolution results from the application of a downscaling method that combines highly accurate, but low-resolution SMOS radiometric information (SMOS L2 data) with high-resolution (brightness temperature measurements), but low sensitivity, visible-to-infrared imagery (MODIS) to SSM across spatial scales (Piles et al 2010, 2014; Sanchez-Ruiz et al. 2014). Brightness temperature measurements from SMOS were combined with NDVI (Normalized Difference vegetation Index) and LST (Land Surface Temperature) from Aqua MODIS. Since MODIS does not measure under cloudy conditions, the SMOS-L4$^{2.0}$ product was affected by the presence of clouds. In the new version 3.0, ERA-Interim LST data is introduced in the MODIS LST/NDVI space, thus, providing soil moisture measurements independently of the cloud conditions. ERA-Interim provides a resolution of about 0.125°, whereas MODIS is a ~ 1 km product. This information has been added in section 2.2.

**2) Another concern with the use of ERA-Interim LST data for downscaling SMOS data. As the LST is derived numerically from the ERA-Interim soil moisture data via the energy budget model of TESSEL, would it be equivalent to use the ERA-Interim soil moisture data directly?**

The methodology used to derive the SMOS-L4 2.0 and 3.0 products has been developed at the Barcelona Expertise Center (BEC). All references provided in this manuscript define the methodology followed and present the results obtained by the multiple validation exercises performed evidencing the quality of the data and supporting the use of the 3.0 product, as we do in this investigation. We are just users of these products and it is out of our scope and the scope of this paper to discuss the methodology applied for the derivation of the products. In any case, we understand that ERA-Interim LST data are used just to extend the downscaling SMOS L4 data to all weather conditions.

**3) Evaluation of the SSM product:**
**Line 366: "the higher resolution SMOS L4 showing lower standard deviation".**

**Line 415: "The CVs of the spatially averaged SMOS L4 is lower than those of SMOS L3 and L2 and in situ observations indicating that this data are less scattered."**
**In my opinion, a lower variability for the downscaled SSM product is unexpected. It should be the opposite: higher variability for the downscaled SSM.**

In lines 368 to 384, we describe the reasons behind the lower variability obtained when temporal means (seasonal) of SMOS L4 are evaluated, which is in relation with the limited temporal availability of the product dictated by the revisit period of the satellite. Furthermore, in the new version 3.0 the use of the coarse resolution ERA-Interim LST in the high-resolution MODIS LST/NDVI space to provide soil moisture measurements independently of the cloud conditions could explain the reduced spatial variability of the SMOS L4 3.0 soil moisture product.

In lines 411 to 415, we discuss that the averaged SMOS-L2 and -L4 3.0 data over the IP are much more variable than the SMOS-L3, showing a more extreme daily index (SMOS-L2: -1 to 2; SMOS-L43.0: -0.7 to 412 1.45). Over the VAS, SMOS-L2 is more variable than the higher resolution SMOS-L4 3.0. But, the last one shows a wider range of values as well as more extreme daily index values when compared to the averaged in situ soil moisture measurements.

**Line 393: "L4 product shows SSM mean and variability in the same range of the SMOS L2 and L3 products, but with a finer-improved resolution representation of the spatial distribution".**
**L398: "the potential added value of the 1 km product is manifest".**
**The SMOS L4 has a spatial variability much lower than that of both L2 and L3 products. How to demonstrate that the slight 1 km variability is real information and not an artefact (oversampling)?**

In lines 395 to 398, we discuss that at sub-seasonal (event) scales "comparisons with the mean ground-based SSM at the VAS (OBS area: *0.25 ±0.0002*) show better agreement with the mean SSM from the SMOS-L4 3.0-1 km disaggregated product (*0.23±0.002*) and poorer correlation with SMOS-L2 (*0.20±0.002*). The problematic of SMOS-L4 3.0 on seasonal time scales vanishes at sub-seasonal (event) scales where the potential added value of the 1 km product is manifest."

Individual comparisons with single in situ measurements from the VAS network (covering a 10 x 10 km$^2$ area with a temporal resolution of 10 min) reveal correlation coefficients higher than 0.7 (e.g. Table 3, Figure 7 and 8).

**Line 633: "consistent with the finer resolution of this product which better captures local information on the 1 km x 1 km pixel, whereas coarser products smooth out this vital information".**
**To me, there is no information in this paper supporting the hypothesis that the downscaled product improves the spatial representation of SMOS L2 and L3 products. To really evaluate the SMOS L4 product, one should compare (in Table 3 for instance) the SMOS L4 versus in situ and SMOS L2 (or L3) versus in situ for each station separately, that is at a scale finer than the L2/L3 spatial resolution. Are statistics better for L4 than for L2 or L3?**

**Bottom sub-table of Table 3 is unclear. In addition errors are identified in the right column (OBS), which does not always correspond to the mean for all stations (?).**

The spatio-temporal correlations are analysed through comparison with point-scale observations over the VAS region. Section 4.2, lines 438 to 477, is devoted to the comparison of SMOS L4 and –L2 products to the in situ measurements from the VAS network. Statistics for individual comparisons at all stations are summarized in Table 3. Figures 7, 8 and even 9 are devoted to these comparisons, although it is not possible to always show all stations due to space issues. In the description, details are given about the better accuracy of the –L4 product. Comparisons with -L3 product are similarly performed but no included in the manuscript because of space issues and not significant results. But following the reviewer suggestion we have included in this section the following paragraph: "Comparisons between SMOS-L3 and ground measurements were similarly performed evidencing the expected bad correlations ($R^2 \sim 0,002$, not shown)".

The legend in Table 3 has been improved to better the reader's understanding about the information provided. The names of the individual stations in the VAS network have been defined for clarification. We have explained relevant calculation methodologies and the content of the table. Also errors in the OBS column have been corrected.

**"Table 3:** Statistics of the comparisons between SMOS-L2 and SMOS-L4$^{3.0}$ soil moisture versus ground-based measurements in the VAS network (the area covering the ground-based network has been called OBS, Figure 1). SMOS descendent orbits are selected for the comparison. Characteristics of the individual stations are given in Table 1. The acronyms for the names of the stations are as follows: (M-I: Melbex_I, M_II: Melbex_II, VAS: VAS, NIC: Nicolas, EZ: Ezpeleta, LC: La Cubera). The period December 2011 to December 2012 is evaluated. The seasonal analysis follows the hydrological cycle. OBS stands for the average of (i) SMOS-L2 and/or SMOS-L4$^{3.0}$ soil moisture values within the $10 \times 10$ km$^2$ where the ground-based network is placed, and (ii) in the case of the in situ observations it refers to the mean of all stations. In Table (a) a seasonal comparison between the mean of all in situ stations and the corresponding mean of SMOS-L2 and/or SMOS-L4$^{3.0}$ soil moisture values within the $10 \times 10$ km$^2$ area is presented. In (b) SMOS-L2 and SMOS-L4$^{3.0}$ soil moisture observations are compared to point-like ground measurements using the closest grid point. The column on the right shows the mean of all stations."

**4) In the present form, the paper is a bit lengthy. The description of approaches is sometimes repetitive. The structure of the manuscript could be improved. For instance: lines 334-335 (and lines 507 to 512) three to four initialization experiments are presented, but the initialization using SMOS data is not mentioned, although claimed as the main objective of the paper. Conclusions are confusing as well. The authors should better highlight their findings by selecting few key results.**
The objective with the different initialization experiments described in lines 334-335 was to demonstrate the impact of initialization on the simulation of SSM. Commonly used initialization values are employed in this perturbation experiment to assess the consequent variability that could be expected in the evolution of the simulated SSM. In lines 340 to 344, the experiments using SMOS L4 3.0 for initialization are introduced.
This part will be reduced and improved to better reflect our purposes. Conclusions will be also rewritten to highlight our findings instead of summarizing our results.

**5) As the study focuses on SMOS derived SSM at high spatial temporal resolution including all weather conditions, I suggest two recent references to complement the state-of-the-art presented in the introduction:**

**Malbéteau, Y., Merlin, O., Balsamo, G., Er-Raki, S., Khabba, S.,Walker, J. P., Jarlan, L. (2018). Toward a Surface Soil Moisture Product at High Spatiotemporal Resolution: Temporally Interpolated, Spatially Disaggregated SMOS Data. Journal of Hydrometeorology, 19(1), 183-200.**

**Djamai, N., Magagi,R., Goïta, K., Merlin, O., Kerr, Y., Roy, A. (2016). A combination of DISPATCH downscaling algorithm with CLASS land surface scheme for soil moisture estimation at fine scale during cloudy days. Remote Sensing of Environment, 184, 1-14.**

Thank you for the additional references both will be included in the manuscript. This will also give us the opportunity to point out relevant differences between the investigated products:

"Recently, complementary studies have produced similar high-resolution SMOS-L4 products such as those of Malbéteau, Y., et al (2018) and Djamai, N., et al (2016). Being similar, however, the algorithms originating them are totally different from those of SMOS-L4$^{3.0}$ used in our study. Whereas SMOS-L4$^{3.0}$ products proceed from the original SMOS-L2 (15 km resolution soil moisture) disaggregated by 1-km MODIS LST and NDVI and modulated with 0.125°-resolution ERA-Interim LST for all-weather conditions, Malbéteau, Y., et al (2018) and Djamai, N., et al (2016) products proceed from the original SMOS-L1 (15 km resolution brightness temperature)."

**6) Line 529: "soil moisture initialization in spatialized SURFEX simulations requires a single representative value for the whole simulation area. In this case, we use as input the SMOS L4 1 km disaggregated soil moisture mean over the whole simulation area for the initialization day". Why not initializing the model at 1 km resolution if 1 km resolution data are available? What is the point of disaggregating SMOS L2/L3 data then?**

The approach proposed by the reviewer would be the ideal to demonstrate the potential of the SMOS L4 3.0 product. However, this is not possible with the SURFEX-ISBA model which requires a single representative soil moisture value for the simulations. We wanted to demonstrate that even when a single upscaled value is used results better reflect the evolution of SSM.

In a new study of the first author, which is about to be submitted to HESS, the suggestion of the reviewer is explored, in which we assess the benefit of using the SMOS-L4 product for the initialization of high-resolution convective-permitting simulations to improve the predictability of extreme weather phenomena such as heavy precipitation.

**7) On the usefulness of surface soil moisture data to initialize ISBA. Line 229: "Particularly relevant for this study is the specific definition of the soil hydraulic parameters which they made for the VAS area, since most of the hydrological parameters are site dependent". Does the approach require in situ measurements for the calibration? Since the objective is to initialize ISBA using SMOS L4 data, I am wondering whether the site specific calibration could be done using SMOS L4 data solely (without relying on in situ measurements for ISBA simulations).**

For the initialization of the model additional soil information, namely, texture (silt, sand and clay percentages), runoff, root-zone soil moisture and other hydraulic parameters in addition to SSM are needed, and those are not provided by SMOS. Most of these parameters were taken from a previous study carried out over the same area (Juglea et al. 2010a and b)

Juglea, S., Kerr, Y., Mialon, A., Wigneron, J.-P., Lopez-Baeza, E., Cano, A., Albitar, A., Millan-Scheiding, C., Carmen Antolin, M., and Delwart, S.: Modelling soil moisture at SMOS scale by use of a SVAT model over the Valencia Anchor Station (2010a). Hydrol. Earth Syst. Sci., 14, 831–846, doi:10.5194/hess-14-831-2010

Juglea, S., Y. Kerr, A. Mialon, E. Lopez-Baeza, D. Braithwaite, and K. Hsu (2010b). Soil moisture modelling of a SMOS pixel: interest of using the PERSIANN database over the Valencia Anchor Station. Hydrol. Earth Syst. Sci., 14, 1509–1525, doi:10.5194/hess-14-1509-2010

**Line 488: "Initialization of land surface models is a crucial issue and its impact on the accuracy of model estimation is widely recognized to be significant". What about the initialization of the root-zone soil moisture, which has supposedly more weight in the initialization than the SSM?**

As above described, root-zone soil moisture has been used from previous studies/observations in the area (Juglea et al., 2010a), however, we did not used this variable in our analysis since SMOS only provides ~ 3-5 com SSM. We included this information in the paper for clarification.

**Specific points:**
**- It is unclear at which spatial resolution ISBA model is run over the VAS?**
The simulations are at 1 km resolution. This has been better clarified in the text.

**- Confusion is often made between observation and sampling grid resolution. Ex. Line 10: 25 km and 15 km are the resolutions of sampling grids, the actual spatial resolution for both products being about 40 km.**
This will be properly clarified in the text.

**- Figure 2 (and Figure 4): Image at the middle is not correctly georeferenced compared to the left (top) and right (bottom) images.**
This has been corrected

**- Units in m3/m3 are sometimes missing the text and the figures.**
This will be corrected

**- Line 306: "SMOS L4 soil moisture grid cells are averaged over the 10x10 km2 area and compared to the mean from the soil moisture network stations to address the issue related to spatial averaging". Please clarify the issue to be addressed?**

Due to the high spatial and temporal variability of the upper 5 cm SSM the sampling of observations is a critical issue. We perform comparison between SMOS and in situ measurements at single locations/stations as well as using the averaged values over the area covered to address this issue.

**- Notations: SURFEX-SAFRAN (SURFEX forced by SAFRAN), SURFEX-ECMWF (SURFEX forced by ERA-Interim) and SURFEX-ISBA are used. The terminology SURFEX-ISBA is confusing as it corresponds to SURFEX (ISBA) forced by station based meteorological measurements. For clarity, I suggest to replace SURFEX-ISBA by (for instance) SURFEX-VAS**

This could be modified for clarity. We propose SURFEX (ISBA) instead

**- Some references are missing in the reference list: I have noted Louvet et al. 2015; Piles et al. 2015; and maybe others.**

The list of references has been revised and necessary corrections have been made.

---

## Author Response (AR2)

**Answer to reviewer 2**

**The authors partly addressed the reviewers' comments. In my opinion, the main purpose of the paper, i.e., showing the improved performance of the all-weather SMOS L4_3.0 soil moisture product is not achieved. As mentioned in the HESSD review, I do not think that by performing a temporal analysis the added-value of a downscaled product can be evaluated. A spatial analysis should be performed. Looking at some of the figures, it seems that the proposed SMOS product fails in reproducing high variability of soil moisture at 1 km scale. Actually, it is not shown explicitly in the paper. Also the analysis for the initialization of SURFEX is weak, and not clear to me.**

**However, I do not want to force the authors to follow my suggestions. Several paper showing a downscaled soil moisture product have been accepted (in my opinion not correctly), therefore I may be wrong. Without a detailed spatial analysis, we must not say that the downslcaed product are representative of high resolution. It is valid for all products, and also for modelling.**

We agree with the reviewer that both a spatial and a temporal analysis are necessary to accurately describe the quality and added value of a downscaled product. We also agree that a detailed spatial analysis is needed to show that a downscaled product accurately represents high resolution. As we described previously in our answers to the reviewer, the temporal and spatial accuracy of the downscaled 1 km product has been previously extensively investigated and proved in many studies some of them referenced in this manuscript (e.g. Piles et al. 2011, 2013, 2014, 2015; Djamai et al 2016; Malbéteau et al 2018). Different observational networks around the world with different number of stations, positions, regional characteristic etc.. were used for this purpose.

In this study, we make use of an additional network with different characteristics to complement previous studies and illustrate our analysis. In agreement with all previous publications we used the information from the several in situ stations that form the VAS soil moisture network to assess the temporal as well as the spatial representativity of the downscaled product. In our analysis we pointed out the advantages and disadvantages of this product considering different temporal and spatial scale. Unfortunately, we are all constrained by the limitations in the number of stations, their locations, distance, soil characteristics etc… which come given mostly by a previous experimental campaign. We do not believe that any soil moisture product should be validated against model simulations, not even at very high resolution, since it is known that model simulations present significant biases in this regard which have to be considered and corrected.

We agree this is a relevant point raised by the reviewer, therefore, we propose to include in the conclusion the following paragraph:

*" This study also points out that in order to more accurately examine the reproducibility of the high spatial variability of this variable by the newly available satellite derived downscaled high-resolution soil moisture observations, large and dense networks of in situ soil moisture measurements covering different soil types and land uses as well as considering different soil depths are needed. In an effort to come a step forward in this direction, dedicated long-term*

*networks with the previously described characteristics should be established permanently in different regions around the world."*

Regarding the analysis for the initialization of SURFEX it has been indicated in the text (section 4.3.2) in agreement with the suggestions of the reviewers that this has to be just considered as an exercise to illustrate the relevance of the issue

[revised manuscript text omitted]